# Dynamics of diffusive cell signaling relays

**Paul B Dieterle[1], Jiseon Min[2], Daniel Irimia[3], Ariel Amir[4]***

[1]Department of Physics, Harvard University, Cambridge, United States; [2]Department of Molecular and Cellular Biology, Harvard University, Cambridge, United States; [3]BioMEMS Resource Center and Center for Surgery, Innovation and Bioengineering, Department of Surgery, Massachusetts General Hospital, Boston, United States; [4]John A. Paulson School of Engineering and Applied Sciences, Harvard University, Cambridge, United States

**Abstract** In biological contexts as diverse as development, apoptosis, and synthetic microbial consortia, collections of cells or subcellular components have been shown to overcome the slow signaling speed of simple diffusion by utilizing diffusive relays, in which the presence of one type of diffusible signaling molecule triggers participation in the emission of the same type of molecule. This collective effect gives rise to fast-traveling diffusive waves. Here, in the context of cell signaling, we show that system dimensionality – the shape of the extracellular medium and the distribution of cells within it – can dramatically affect the wave dynamics, but that these dynamics are insensitive to details of cellular activation. As an example, we show that neutrophil swarming experiments exhibit dynamical signatures consistent with the proposed signaling motif. We further show that cell signaling relays generate much steeper concentration profiles than does simple diffusion, which may facilitate neutrophil chemotaxis.

*For correspondence:
arielamir@seas.harvard.edu

**Competing interests:** The authors declare that no competing interests exist.

## Introduction

Prototypical diffusive signaling – in which individual cells communicate with neighbors by releasing diffusible molecules into the extracellular medium – is a relatively slow process. Signaling molecules undergoing random walks in the extracellular medium have a root mean square displacement that grows like the square root of both the time since emission, $t$, and the signaling molecule diffusivity, $D$. It follows that the distance an individual cell can signal also grows like the square root of time. Thus, for thousands of cells coordinating actions over millimeters, simple diffusive signaling with small molecules ($D \approx 10^{-10}$ m$^2$/s) takes hours. These length and times scales are incommensurate with observed behavior in developmental biology (*Chang and Ferrell, 2013*; *Cheng and Ferrell, 2018*; *Vergassola et al., 2018*), immune response (*Reátegui et al., 2017*), and microbial consortia (*Parkin and Murray, 2018*), in which cells exchanging diffusible molecules coordinate activity over millimeters in tens of minutes.

Indeed, when many cells collectively integrate environmental cues and participate in the signaling, they can propagate diffusive waves with a fixed speed, $v$, in the asymptotic limit. This effect and its analogs have long been studied in the context of excitable media (*Keener and Sneyd, 2009*; *Keener, 1987*; *Muratov, 2000*) and observed in biological phenomena as diverse as natural cell signaling circuits (*Noorbakhsh et al., 2015*; *Pálsson and Cox, 1996*; *Kessler and Levine, 1993*; *Gelens et al., 2014*), synthetic cell signaling circuits (*Parkin and Murray, 2018*), apoptosis (*Cheng and Ferrell, 2018*), range expansions (*Tanaka et al., 2017*; *Fisher, 1937*; *Kolmogorov et al., 1937*; *Barton and Turelli, 2011*; *Gandhi et al., 2016*; *Birzu et al., 2018*), and development (*Chang and Ferrell, 2013*; *Vergassola et al., 2018*; *Muratov and Shvartsman, 2004*; *Nolet et al., 2020*). In this way, small groups of cells can transmit signals more quickly than simple diffusion allows by recruiting the help of their neighbors.

While diffusive waves have been observed in a variety of biological processes, they have also been experimentally probed in a variety of spatial contexts – in quasi-1D tubes (*Cheng and Ferrell, 2018*; *Chang and Ferrell, 2013*; *Nolet et al., 2020*), in quasi-2D droplets and chambers (*Nolet et al., 2020*; *Afanzar et al., 2020*), on 2D surfaces in fly eggs (*Vergassola et al., 2018*), and on substrates of finite thickness (*Parkin and Murray, 2018*; *Pálsson and Cox, 1996*). And while the phenomenology of diffusive waves has been studied for years, in the context of cell signaling it is less well-understood how the propagation and initiation of such waves are affected by the dimensionalities of the cellular distribution and the diffusive environment – or even how to identify the system dimensionality – as previous modeling work has largely assumed quasi-1D dynamics (*Kessler and Levine, 1993*; *Meyer, 1991*; *Gelens et al., 2014*; *Vergassola et al., 2018*). Also unclear is how robust the resulting signaling dynamics are to underlying biological details, such as the shape of the function governing cell activation and signaling molecule emission.

Here, we revisit the propagation and initiation of diffusive waves in the context of cell signaling. Through a comprehensive study of single-component relays — in which cells measure the local concentration of a signaling molecule and participate in the emission of the same molecule — we show that the asymptotic wave dynamics of diffusive relays are governed by simple scaling laws. In some system dimensionalities, these scaling laws are identical to famous results from the 20th century (*Fisher, 1937*; *Kolmogorov et al., 1937*; *Luther, 1906*); in other system dimensionalities, we show that these well-known scaling laws can be drastically altered. For example, cells confined to two (or one) dimensions with signaling molecule diffusion in three (or two) dimensions give rise to a diffusive wave whose speed has no dependence on $D$: a wave driven by diffusion whose speed does not depend on the rate of diffusion. In contrast to the dramatic effect of system dimensionality, these scaling laws are insensitive to many biological details, including the functional form of cellular activation — the dependence of signaling molecule emission rate on the local concentration. We additionally account for other phenomena – molecule decay, pulsed emission, and the discreteness of cells – that *do* affect the asymptotic wave dynamics; in so doing, we provide an intuitive rubric for determining under what conditions these effects alter the wave propagation speed.

In our studies of wave initiation, we systematically examine under what conditions a group of cells can trigger the formation of a diffusive wave. Here again, our results provide predictive relationships between biophysical inputs and the resulting dynamics, which are at once dramatically affected by dimensionality and largely insensitive to the details of activation and cellular uptake.

Finally, we show that neutrophil swarming experiments (*Reátegui et al., 2017*) display dynamics consistent with our model. In this context, our results elucidate a potential design principle of diffusive relays: they create large concentration gradients. Whereas simple diffusion of a signaling molecule from a central source creates a shallow concentration profile that falls off like $\exp(-r^2/4Dt)$, relays give rise to steep concentration profiles with gradients that quickly propagate outward and decay only modestly inside the wave front. As such, for cells like neutrophils – which use a small molecule, leukotriene B4 (LTB4), as an intercellular signaling molecule and chemoattractant (4, 18, 19) – relays may provide a method for cells to collectively generate large, continous chemical gradients that may serve to guide directional migration; the continuous gradients generated by single-component relays contrast with the pulse trains of chemotactic cues observed in, for example, *Dictyostelium discoideum* (*Kessler and Levine, 1993*; *Pálsson and Cox, 1996*).

## Results

### Model construction

We begin by considering a static group of cells uniformly distributed in two dimensions – for example, atop a solid surface – and described by an area density $\rho$ (*Figure 1A*). We assume a cell at position $\mathbf{r}$ senses the local concentration of a signaling molecule, $c(\mathbf{r}, t)$, and participates in the emission at a concentration-dependent rate $af(c)$ with $a$ the maximum rate and $f(c)$ a dimensionless function. Once secreted into the extracellular medium, the signaling molecules diffuse with diffusivity $D$. Treating the cells and signaling molecule concentration in the continuum limit – we discuss the validity of doing in the next section and in Appendix 6: Assessing the validity of a continuum analysis – gives rise to a single equation that governs the time evolution of $c(\mathbf{r}, t)$:

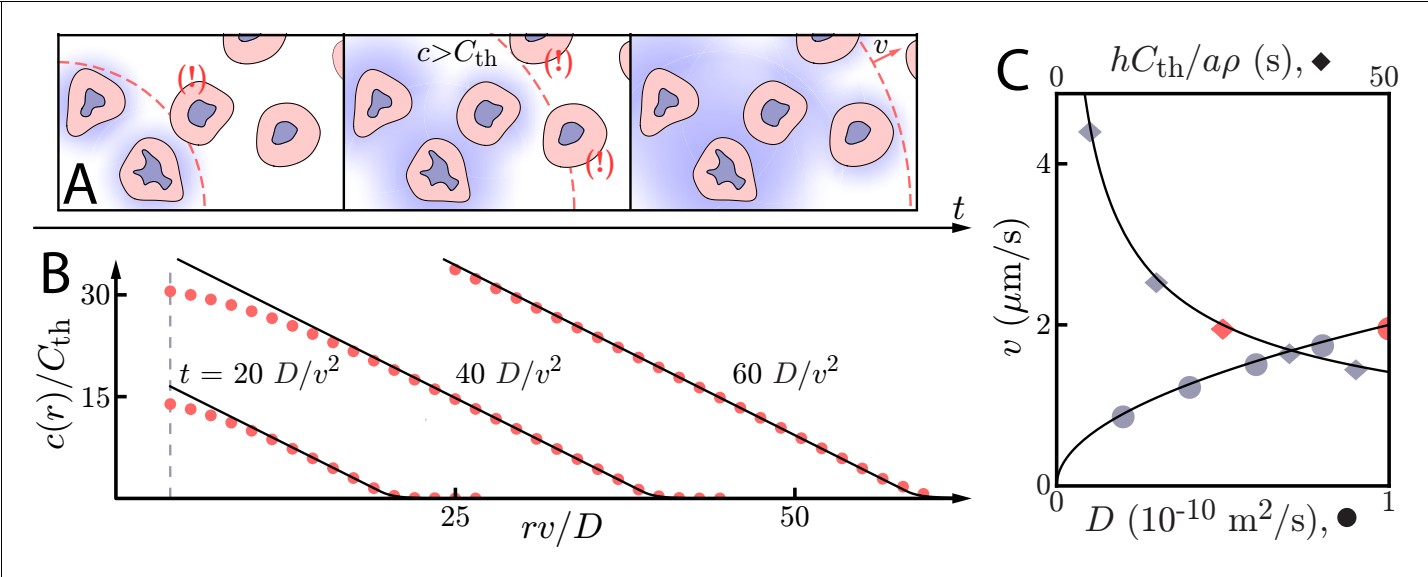

**Figure 1.** Asymptotic relay dynamics of cells in 2D with diffusion in 2D. (**A**) Schematic illustrating the diffusive relay motif. Cells (pink with purple nucleus) release a signaling molecule that diffuses (blue clouds). They do so when the local concentration exceeds a threshold, $C_{th}$. This gives rise to a diffusive wave with wave speed $v$. (**B**) Snapshot concentration profiles. Asymptotic theory (**Equation (6)**, black lines) and numerical simulation of **Equation (2)** (red dots, details of the numerical methods can be found in Materials and methods) are in good agreement and show outward-propagating waves. Here, $D = 10^{-10}$ m$^2$/s, $v = 2$ µm/s, and $hC_{th}/a\rho = D/v^2$. Numerical simulations assume that a cell colony of size $r_i = 4D/v$ (dashed vertical line) centered at the origin starts signaling $t = 0$. (**C**) Numerical wave speed as measured at $t = 100D/v^2$ (markers) agrees well with theory (**Equation (5)**, black line) as we independently vary $D$ (circles) and $hC_{th}/a\rho$ (diamonds) relative to the panel B values (red circle and diamond).

$$\frac{\partial c}{\partial t} = D\nabla^2 c + a\rho\delta(z)f(c) \tag{1}$$

where the Dirac delta function $\delta(z)$ accounts for the fact that the cells are confined to the plane. The source function $f(c)$ is in general a complicated non-linear function of $c$. It can include uptake, release, and cell-induced degradation of the signaling molecule – or any other process proportional to the local cell density. We will consider this general case shortly. To start, we consider a simple case in which cells measure the local signaling molecule concentration, $c$, and participate in the emission only if $c$ exceeds a threshold concentration, $C_{th}$. In such a case, the activation function $f(c)$ is well described by a Heaviside step function $\Theta[c - C_{th}]$ and the concentration dynamics obey

$$\frac{\partial c}{\partial t} = D\nabla^2 c + a\rho\delta(z)\Theta[c - C_{th}]. \tag{2}$$

Additionally, while we at first consider cells scattered in a two-dimensional plane, one can study the signaling dynamics of cells in a one-dimensional channel or a three-dimensional environment with similar analyses. Below, we discuss the connections between the cell signaling dynamics in all these scenarios, and all are treated in depth in Appendix 2: Asymptotic wave ansatz.

## Asymptotic wave dynamics

Our first step in understanding diffusive signaling relays is to solve for the asymptotic dynamics of **Equation (2)**. Since such relays involve cells signaling their neighbors, which then signal their own neighbors, one can imagine that diffusive relays give rise to diffusive waves. We therefore make the ansatz that the concentration $c(\mathbf{r}, t) = c(r, z, t)$ can be described by an outward-traveling wave of the form $c(r, z, t) = c(\tilde{r} = r - vt, z)$ (**Fisher, 1937**; **Kolmogorov et al., 1937**; **Tanaka et al., 2017**). Here, $\tilde{r}$ is the distance from the wave front – negative when inside the wave front, positive when beyond – and $v$ is the wave speed. In essence, we wish to examine the wave from the perspective of an observer moving at the wave front. With $C_{th} \equiv c(\tilde{r} = 0, z = 0)$ and $r \gg D/v$, we take **Equation (2)** and arrive at the following equation governing asymptotic behavior:

$$0 = D\left(\frac{\partial^2 c}{\partial \tilde{r}^2} + \frac{1}{r}\frac{\partial c}{\partial \tilde{r}} + \frac{\partial^2 c}{\partial z^2}\right) + v\frac{\partial c}{\partial \tilde{r}} + a\rho\delta(z)\Theta[c - C_{\text{th}}]$$

$$\approx D\left(\frac{\partial^2 c}{\partial \tilde{r}^2} + \frac{\partial^2 c}{\partial z^2}\right) + v\frac{\partial c}{\partial \tilde{r}} + a\rho\delta(z)\Theta[c - C_{\text{th}}] \qquad (3)$$

$$= D\left(\frac{\partial^2 c}{\partial \tilde{r}^2} + \frac{\partial^2 c}{\partial z^2}\right) + v\frac{\partial c}{\partial \tilde{r}} + a\rho\delta(z)\Theta[-\tilde{r}].$$

Since we consider $r \gg D/v$, we may ignore the $D(\partial c/\partial \tilde{r})/r$ term due to the dominance of $v\partial c/\partial \tilde{r}$. This is effectively the same as ignoring the curvature of the wave front and has the effect of reducing our asymptotic analysis of cells in two dimensions into an asymptotic analysis of cells in one dimension (*Tanaka et al., 2017*). The asymptotic dynamics of cells distributed in three spatial dimensions allow for a similar manipulation (see Appendix 2: Asymptotic wave ansatz).

We wish to find a solution to *Equation (3)* for various diffusive – that is, extracellular – environments. In doing so, we hope to solve for the spatial dependence of the concentration profiles $c(\tilde{r}, z)$ as well as a relationship that will tell us how the signaling dynamics – in this case, the wave speed $v$ – depend on the biophysical system parameters like the cell density, $\rho$; the concentration threshold, $C_{\text{th}}$; and the signaling molecule emission rate, $a$.

But first, we note that *Equation (3)* provides two quantities of value: a natural length scale $D/v$ and a natural time scale $D/v^2$. For a small diffusing molecule with $D \approx 10^{-10}$ m$^2$/s and a wave speed of $v \approx 1$ µm/s – approximately the numbers relevant for several experimental systems (*Cheng and Ferrell, 2018*; *Chang and Ferrell, 2013*; *Parkin and Murray, 2018*; *Vergassola et al., 2018*; *Pálsson and Cox, 1996*) including, as we show below, neutrophil swarming (*Reátegui et al., 2017*) – we recover $D/v \approx 100$ µm and $D/v^2 \approx 100$ s. We have already used the natural length scale $D/v$ to derive *Equation (3)* and to show that cells in 2D have the same asymptotic dynamics as cells in 1D or 3D, and we can use these scales to further justify several other approximations we have made so far. For instance, the approximation that the out-of-plane cell density can be described by $\delta(z)$ is valid when the cell size $H \ll D/v$; similarly, decay of the signaling molecule can be neglected for a decay rate $\gamma \ll (D/v^2)^{-1}$ while pulsed emission gives rise to the same asymptotic wave speed if the width of the pulse $\tau$ satisfies $\tau \gg D/v^2$. Finally, we note that the use of *Equation (2)* as a starting point is justified when the mean distance $d$ between neighboring cells satisfies $dv/4D \ll 1$. A thorough, mathematical discussion of all the above, including a demonstration of why $D/v$ and $D/v^2$ are the appropriate scales, is presented in Appendix 2: Asymptotic wave ansatz.

When the extracellular medium thickness $h \ll D/v$, diffusion of the signaling molecule is effectively two-dimensional as we can take $\partial^2 c/\partial z^2 \to 0$ and $\delta(z) \to 1/h$. In this limit, *Equation (3)* becomes

$$h \ll D/v : 0 \quad = D\frac{\partial^2 c}{\partial \tilde{r}^2} + v\frac{\partial c}{\partial \tilde{r}} + \frac{a\rho}{h}\Theta[c - C_{\text{th}}]$$

$$= D\frac{\partial^2 c}{\partial \tilde{r}^2} + v\frac{\partial c}{\partial \tilde{r}} + \frac{a\rho}{h}\Theta[-\tilde{r}] \qquad (4)$$

which we can solve to find the asymptotic dynamics of cells in 2D (1D, 3D) with effective signaling molecule diffusion in 2D (1D, 3D) – the thin extracellular medium limit (*Figure 1*). This corresponds to the long-pulse, long-decay time limit of the model constructed by *Kessler and Levine, 1993* and is similar to the model considered by *Meyer, 1991*. Adding signaling molecule decay to *Equation (4)* would yield a model first considered by *McKean, 1970* in the context of nerve impulse propagation.

Before solving *Equation (4)* exactly, we make two crucial observations from which we can derive the functional form of the wave speed, $v$. First, because the source (furthest right) term in *Equation (4)* is proportional to $a\rho/h$, all concentrations in the problem, including $C_{\text{th}}$, are proportional to $a\rho/h$. As the non-source terms in *Equations (4) and (1)* are linear, the only role $a\rho/h$ serves is to set the concentration scale of the dynamics. Thus, $C_{\text{th}}$, $a$, $\rho$, and $h$ combine to give us a single model parameter to describe the threshold concentration, $hC_{\text{th}}/a\rho$, which has units of time (measured in s). Second, the only other parameter in the problem besides $v$ – which we want to calculate – is the diffusion constant, $D$, which has units of length squared divided by time (measured in m$^2$/s). Thus, the only combination of these two parameters that will give a speed (measured in m/s) is $(a\rho D/hC_{\text{th}})^{1/2}$.

By this simple dimensional analysis argument, the wave speed $v$ can only be $v = \alpha(a\rho D/hC_{\text{th}})^{1/2}$ for some constant $\alpha$. Formally, the above procedure is equivalent to non-dimensionalizing *Equation (4)*, as discussed in Appendix 2: Asymptotic wave ansatz.

By the same reasoning, *any* activation function $f(c)$ – a Heaviside step function, a Hill function, or even a bistable function – that can parameterized by a single concentration $C_{\text{th}}$ and emission rate $a$ must give the same scalings if it has a traveling wave solution. While we focus on positive activation functions in this work, we emphasize that if signaling molecule degradation is dominated by cell-induced processes like uptake, then signaling molecule degradation is also proportional to the cell density and the resulting (presumably bistable) production curve will yield dynamics that are also beholden to this scaling law.

One can confirm this scaling law for Heaviside activation by solving *Equation (4)* for $\tilde{r}>0$ and $\tilde{r}<0$, then matching boundary conditions at $\tilde{r} = 0$. This analysis indeed reveals that

$$h \ll D/v: \ C_{\text{th}} = a\rho D/hv^2 \implies v = \sqrt{a\rho D/hC_{\text{th}}} \tag{5}$$

while

$$h \ll D/v: c(\tilde{r}) = \begin{cases} -a\rho\tilde{r}/hv + a\rho D/hv^2 & \tilde{r} \leq 0 \\ a\rho De^{-\tilde{r}v/D}/hv^2 & \tilde{r} \geq 0. \end{cases} \tag{6}$$

The concentration of signaling molecule thus grows linearly in the distance inside the wave front and decays exponentially in the distance beyond the wave front. We compare numerical simulations of *Equation (2)* (see Materials and methods for details) with the above asymptotic formulae for wave speeds and concentration profiles in *Figure 1B/C*. For $r \gg D/v$, the asymptotic formulae describe well both the concentration profile and the wave speed.

The wave speed relationship given in *Equation (5)* is analogous to the Fisher-Kolmogorov wave speed (*Fisher, 1937*; *Kolmogorov et al., 1937*; *Gelens et al., 2014*) – with $hC_{\text{th}}/a\rho$ replacing the doubling time as the characteristic time scale in the problem – and has been discussed in beautiful previous work (*Gelens et al., 2014*; *Meyer, 1991*), starting with *Luther, 1906*. Amazingly, Luther's formula, which posits the scaling relation $v \sim \sqrt{D}$, holds even in scenarios beyond those considered here; for instance, waves driven by oscillatory activation dynamics – as are relevant for intercellular signaling in *Dictyostelium discoideum* (*Kessler and Levine, 1993*; *Pálsson and Cox, 1996*) and developmental trigger wave propagation (*Gelens et al., 2014*; *Chang and Ferrell, 2013*) – are subject to this same scaling. One can understand this through simple dimensional analysis. These more complex scenarios add signaling molecule decay and a periodically modulated source function to the above model. Thus, to our set of parameters, $D$ (measured in m$^2$/s) and $hC_{\text{th}}/a\rho$ (measured in s), we add a modulation time $\tau$ (measured in seconds) and decay rate $\gamma$ (measured in 1/s). As $D$ is the only parameter involving a length scale, it must be that $v \sim \sqrt{D}$ even in these more complex scenarios.

By way of contrast, *Vergassola et al., 2018* have shown that an unconventional scaling of $v \sim D^{3/4}$ can result from time-dependent dynamics of the source term at the wave front, a phenomenon that breaks our assumption that all cells obey the same time-independent source function $f(c)$. Similarly, as we will now show, the dimensionality of the system can also have a dramatic effect on wave speed scaling laws.

Next, we consider a thick extracellular medium for which $h \gg D/v$. Such a configuration is relevant for signaling in bacterial consortia atop thick, permeable substrates (*Parkin and Murray, 2018*) or anywhere that a lower dimensional tissue abuts a thick and permeable extracellular environment as can be found, for example, in the retina. Here, the signaling molecules can diffuse out of plane (*Figure 2A*). Because the cells sit atop a solid boundary, signaling molecules can only diffuse in the upper half of the plane and the source term in *Equation (3)* acquires a factor of two to account for this boundary condition:

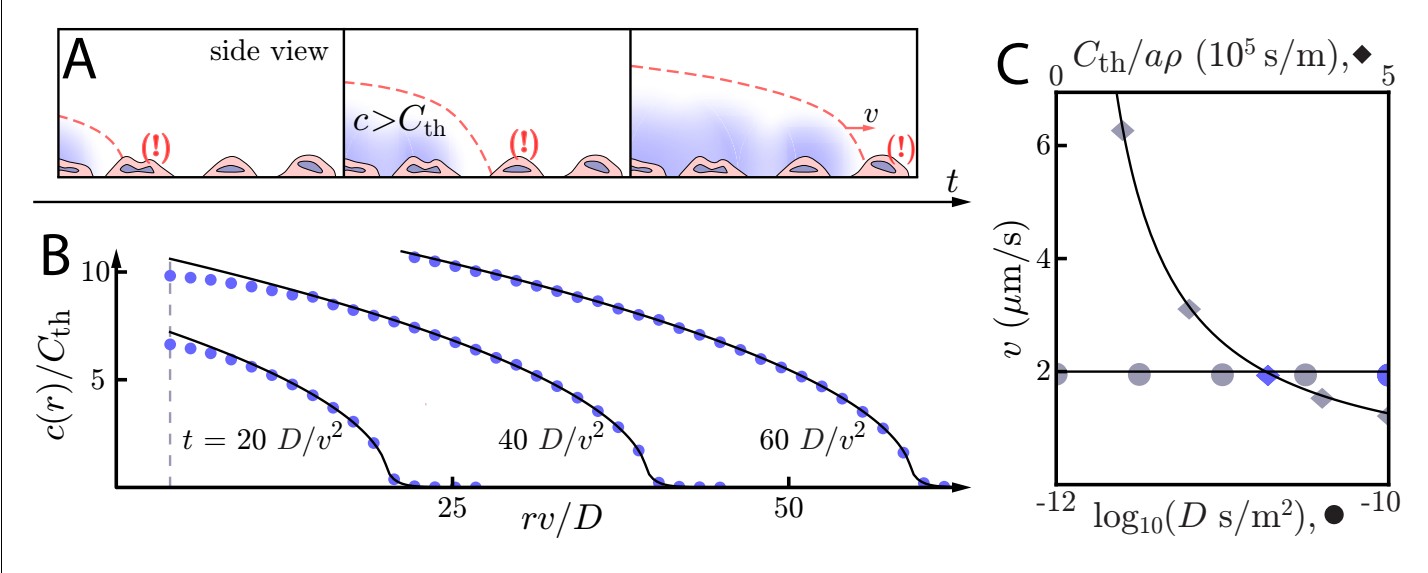

**Figure 2.** Asymptotic relay dynamics with cells in 2D and diffusion in 3D. (A) Schematic of cells (pink with purple nucleus) performing a diffusive relay in which signaling molecules (blue clouds) can diffuse out-of-plane. Here, such relays give rise to a diffusion-constant-independent wave speed, $v$. (B) Snapshot concentration profiles of the signaling molecule show good agreement between numerical simulation of *Equation (2)* (blue dots, details of numerical methods can be found in Materials and methods) and asymptotic theory (*Equation (9)*, black lines). Here, $D = 10^{-10}$ m$^2$/s and $v = 2$ μm/s with $C_{th}/a\rho = 2/\pi v$. The initial signaling colony is of size $r_i = 4D/v$ (dashed vertical line). (C) Numerical wave speed as measured at $t = 100D/v^2$ (markers) agrees well with theory (*Equation (8)*, black line) as we independently vary $D$ (circles) and $C_{th}/a\rho$ (diamonds) relative to the panel B values (blue circle and diamond). As predicted, $v$ is indeed $D$-independent in this system.

$$h \gg D/v : 0 = D\left(\frac{\partial^2 c}{\partial \tilde{r}^2} + \frac{\partial^2 c}{\partial z^2}\right) + v\frac{\partial c}{\partial \tilde{r}} + 2a\rho\delta(z)\Theta[c - C_{th}]$$
$$= D\left(\frac{\partial^2 c}{\partial \tilde{r}^2} + \frac{\partial^2 c}{\partial z^2}\right) + v\frac{\partial c}{\partial \tilde{r}} + 2a\rho\delta(z)\Theta[-\tilde{r}]. \tag{7}$$

Effectively, we have cells in 2D with diffusion in 3D. We note that this case is asymptotically equivalent to cells in 1D emitting into a semi-infinite 2D environment. Thus, comparing to *Equation (6)*, we can see that the asymptotic dynamics are not determined by the dimension of the cell distribution or the diffusive environment, but by the *difference* in dimension between them.

The same dynamics hold for cells on a curved surface (such as epithelia) as long as the length scale of the curvature and the thickness of the extracellular medium are both large compared to $D/v$. If the length scale of the curvature is large compared to $D/v$, but the extracellular medium is thin compared to $D/v$, then the dynamics will be of cells in a 2D plane with diffusion in 2D. Similarly, cells on the surface of a tube with diffusion in the tube's interior will interpolate between these two limits: when the radius of the tube is large compared to $D/v$, the dynamics will be of cells in 2D and diffusion in 3D; when the radius of the tube is small compared to $D/v$, the dynamics will be of diffusion and cells in 1D.

Examining *Equation (7)* as we did *Equation (4)* reveals that every concentration in a thick extracellular medium is proportional to $a\rho$. Thus, we have two independent parameters in *Equation (7)*: $C_{th}/a\rho$ (measured in s/m) and $D$ (m$^2$/s). The only combination of these parameters that will give a wave speed (measured in m/s) is $a\rho/C_{th}$. It therefore *must* be the case that $v = \alpha a\rho/C_{th}$ with $\alpha$ a constant – a wave driven by diffusion whose wave speed is independent of the rate of diffusion. We again stress that this is true for *any* activation function that has a traveling wave solution and can be parameterized by a single concentration $C_{th}$ and a single emission rate $a$. (For Hill function activation, $\alpha \approx 2/\pi$ for $n \geq 2$, see Appendix 5: Asymptotic wave dynamics with Hill function activation.) Thus, the scaling laws governing the asymptotic dynamics are insensitive to the details of single-cell activation.

It is worth reflecting on the fact that some system geometries give a wave whose speed is diffusion constant-independent. This finding implies that, at least in some contexts, the size of the signaling molecule has little to do with the resultant cell signaling speed. We note that this is in contrast with the more standard wave speed scaling in *Equation (5)*, in which smaller (lower molecular weight, higher *D*) signaling molecules result in a faster wave, all else equal.

A full solution of *Equation (7)*, obtained in Appendix 2: Asymptotic wave ansatz by combining a partial Fourier transform in the *z*-dimension and the methods used to solve *Equation (4)*, yields

$$h \gg D/v: \ C_{\text{th}} = 2a\rho/\pi vv = 2a\rho/\pi C_{\text{th}} \tag{8}$$

and

$$h \gg D/v: \ c(\tilde{r}) \approx \begin{cases} 2a\rho(-\tilde{r}/\pi vD)^{1/2} & \tilde{r} \ll -D/v \\ a\rho(D/\pi\tilde{r}v^3)^{1/2}e^{-v\tilde{r}/D} & \tilde{r} \gg D/v. \end{cases} \tag{9}$$

So for cells in a thick extracellular medium, the concentration grows like the square root of the distance inside the wave front and decays exponentially beyond the wave front. As with the 2D diffusive environment, we verify these relationships numerically (*Figure 2B/C*, see Materials and methods for details of the numerical simulations). We see that the wave speed is indeed *D*-independent over two orders of magnitude in the diffusion constant.

The diffusive relay signaling motif therefore gives rise to diffusive information waves for which *Equation (5)* and *Equation (8)* provide predictive relationships between wave speed, threshold concentration, cell density, extracellular medium thickness, and emission rate for a variety of system dimensionalities. Similarly, *Equation (6)* and *Equation (9)* provide quantitative functional predictions of the concentration profiles generated by diffusive relays. By dimensional analysis, these scaling laws are insensitive to the details of activation. Nonetheless, other details – signaling molecule decay, pulsed emission, discreteness of cells – can alter these robust scaling laws (*Keener, 2000*; Dieterle P and Amir A, 2020. Manuscript in preparation). We explicitly discuss these corrections in the appendices (see Appendix 3: Pulsed emission and decay and Appendix 6: Assessing the validity of a continuum analysis), where we also discuss the dynamics of cells in 1D with 3D diffusion and the properties of waves in an arbitrary extracellular medium thickness. Both signaling molecule decay and pulsed emission decrease the steepness of the concentration gradient inside the wave front, and both decrease the wave speed. We emphasize that, in all cases, the asymptotic dynamics are not determined by the dimension of the diffusive or cellular environment, but by the difference in dimension between the two.

## Signaling wave initiation

Armed with a knowledge that diffusive relays birth diffusive waves, we now ask whether such waves are always initiated. As with the asymptotic dynamics, wave initiation depends on the system dimensionality. Here, however, the dimensionality of the diffusive environment alone determines qualitative behavior. Much previous work in chemical waves and excitable media has shown that a delicate interplay of activation, repression, and diffusion can give rise to a host of dimension-dependent wave initiation phenomena (*Foerster et al., 1989*; *Weise and Panfilov, 2011*); our task here is to study the dimension-dependent dynamics of concentration build up in single-component relays.

To begin, we consider an 'initiating colony' of radius $r_i$ in which cells emit a diffusible signaling molecule with rate *a* (*Figure 3A*). The surrounding cells respond by emitting the same signaling molecule according to some activation function, $f(c)$. Here, we take $f(c)$ to be a Hill function of degree *n* (*Figure 3B*).

In one- and two-dimensional diffusive environments, a continuously emitting source leads to a diverging concentration throughout the space. However, in three-dimensional diffusive environments, a continuously emitting source gives rise to a steady-state concentration with $1/r$ tails (*Krapivsky et al., 2010*).

We therefore expect that an initiating colony of cells, regardless of its radius $r_i$ (in fact, even if it consists of a single cell – see Appendix 7: Initiation dynamics), will be able initiate a diffusive wave in one- and two-dimensional environments; meanwhile, a colony in a three-dimensional environment may fail to initiate a diffusive wave. This is indeed what we observe.

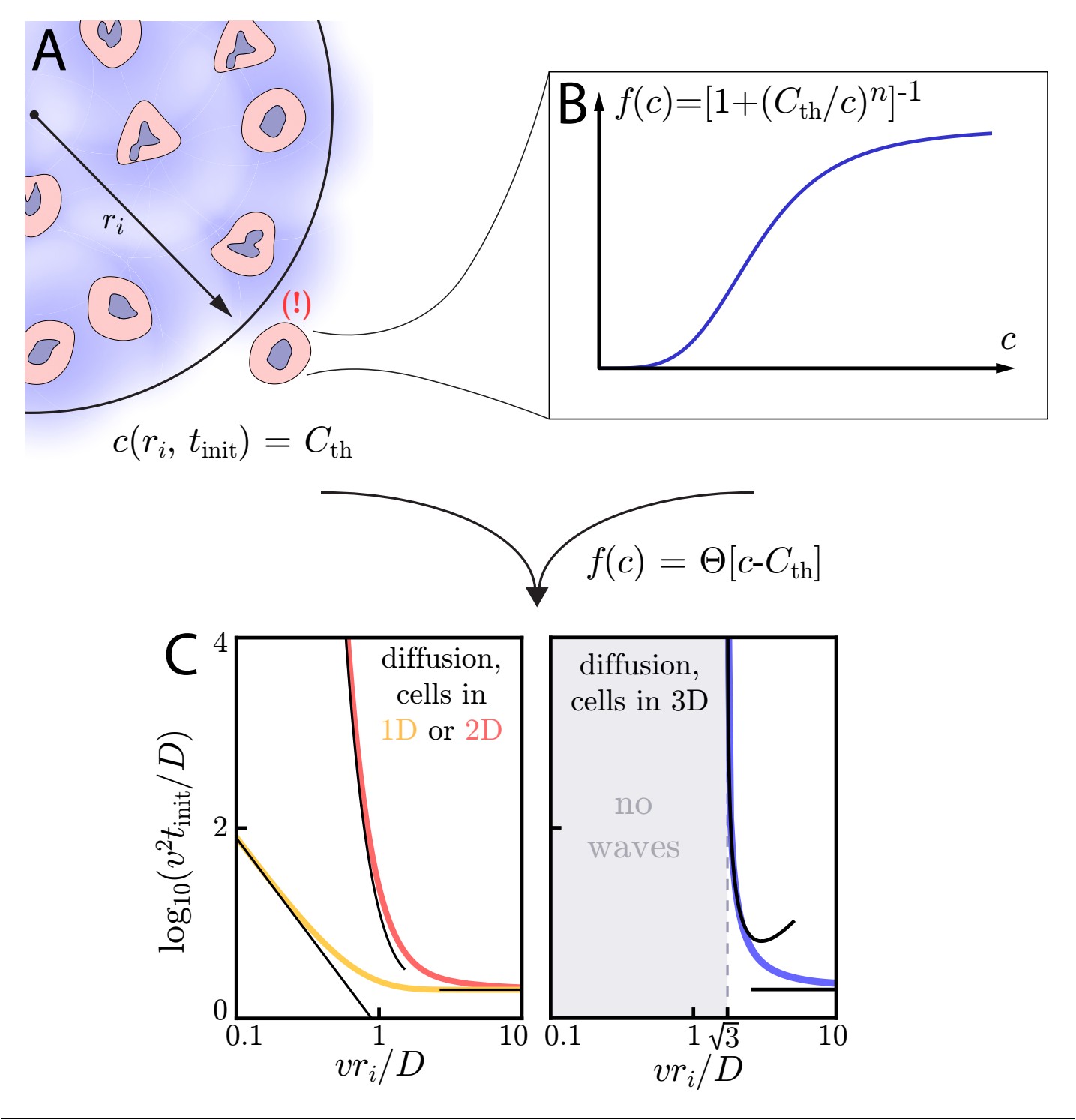

**Figure 3.** Wave initiation dynamics. (**A**) Schematic demonstrating wave initiation. Cells within some initial signaling volume of radius $r_i$ begin signaling at some rate $a$. The signaling wave is initiated when the concentration at nearby cells exceeds the threshold concentration, $C_{th}$. (**B**) Cells near the initial signaling volume participate in the emission according an activation function, $f(c)$. For instance, in the case of Hill function activation, $f(c) = [1 + (C_{th}/c)^n]^{-1}$. C: Initiation times for Heaviside activation, in which $f(c) = \Theta[c - C_{th}]$. Numerics (thick colored lines) and approximate asymptotic theory (**Equations (10) and (11)**, thin black lines) of the initiation time's dependence on $r_i$ for cells and diffusion in 1D or 2D (left) or cells in 2D with diffusion in 3D (right). For cells and diffusion in 1D, **Equation (10)** provides a good approximation in the limits $vr_i/D \ll 1$ and $vr_i/D \gg 1$. Similarly, for cells and diffusion in 2D, **Equation (11)** governs the large and small $vr_i/D$ limits. In both of these cases, the wave always initiates, but the initiation time

*Figure 3 continued on next page*

*Figure 3 continued*

can be orders of magnitude larger than $D/v^2$ if $r_i \ll D/v$. For cells and diffusion in 3D (right), signaling waves do not initiate for $vr_i/D < \sqrt{3}$. Here again, the asymptotic theory **Equation (12)** is in good agreement with numerics.

---

In **Figure 3C**, we demonstrate this dramatic dimension-dependence in the case of switch-like activation, for which $f(c) = \Theta[c - C_{\text{th}}]$. To find the initiation time $t_{\text{init}}$, we integrate Green's functions of the diffusion equation (see Appendix 7: Initiation dynamics for details) to calculate the concentration profile created by cells continuously emitting with rate $a$ inside the initiating colony. When the concentration at $r_i$ is equal to the threshold – $C_{\text{th}} = c(r_i, t_{\text{init}})$ – cells outside the initiating colony begin to participate in the relay and the wave is initiated. Below, we characterize the initiation time as a function of $r_i$ and the characteristic time and length scales – $D/v^2$ and $D/v$, respectively – of a given system, thus linking the initiation dynamics to the asymptotic wave speed, $v$. A summary of these results as a function of dimension, along with a summary of the asymptotic dynamics, can be found in **Table 1**.

For cells in 1D with 1D diffusion, the initiation time in the limits of small ($r_i \ll D/v$) and large ($r_i \gg D/v$) initiating colonies is:

$$t_{\text{init}} \approx \begin{cases} (\pi D/4v^2)(D/vr_i)^2 & r_i \ll D/v \\ t_{\text{min},1,1} = 2D/v^2 & r_i \gg D/v. \end{cases} \tag{10}$$

When $r_i \ll D/v$, the signaling molecules quickly diffuse across and away from the initiating colony. Thus, it is hard for the colony to build up a concentration that exceeds $C_{\text{th}}$. Correspondingly, the initiation time increases like $1/r_i^2$ for small $r_i$. Meanwhile, for $r_i \gg D/v$, the size of the initiating colony becomes irrelevant and reaches a minimum value of $t_{\text{min,1D}}$, determined entirely by the characteristic time scale $D/v^2$. The full dependence of $t_{\text{init}}$ on $r_i$ is pictured in **Figure 3C**, where we show that the above limits are valid approximations.

Next, we consider cells in 2D with diffusion in 2D. Here, for $r_i \ll D/v$, the initiation time scales harshly as

$$t_{\text{init}} \sim \begin{cases} (r_i^2/4D)e^{(2D/vr_i)^2} & r_i \ll D/v \\ t_{\text{min},2,2} = t_{\text{min},1,1} = 2D/v^2 & r_i \gg D/v. \end{cases} \tag{11}$$

---

**Table 1.** Summary of asymptotic and initiation dynamics with Heaviside activation.
For different system dimensionalities, we summarize the asymptotic wave speed, $v$; the initiation time for small initial signaling colony size, $t_{\text{init}}, \frac{vr_i}{D} \ll 1$; and the initiation time for large initial signaling colony size, $t_{\text{init}}, \frac{vr_i}{D} \gg 1$. One-dimensional diffusive environments are assumed to be narrow channels of width $h$ in each direction perpendicular to the channel length. The cell density $\rho$ has units 1/m for cells in 1D, 1/m$^2$ for cells in 2D, and 1/m$^3$ for cells in 3D. When the diffusive and cell dimensions do not match, the environment is assumed to be semi-infinite.

| | $v$ | $t_{\text{init}},$ $\frac{vr_i}{D} \ll 1$ | $t_{\text{init}},$ $\frac{vr_i}{D} \gg 1$ |
|---|---|---|---|
| Cells in 1D, diff. in 1D | $\left(\frac{a\rho D}{h^2 C_{\text{th}}}\right)^{1/2}$ | $\sim \left(\frac{D}{vr_i}\right)^2$ | $2D/v^2$ |
| Cells in 1D, diff. in 2D | $\frac{2a\rho}{\pi h C_{\text{th}}}$ | $\sim \exp\left(\frac{2D}{vr_i}\right)^2$ | $4D/\pi v^2$ |
| Cells in 2D, diff. in 2D | $\left(\frac{a\rho D}{h C_{\text{th}}}\right)^{1/2}$ | $\sim \exp\left(\frac{2D}{vr_i}\right)^2$ | $2D/v^2$ |
| Cells in 2D, diff. in 3D | $\frac{2a\rho}{\pi v}$ | no waves | $4D/\pi v^2$ |
| Cells in 3D, diff. in 3D | $\left(\frac{a\rho D}{C_{\text{th}}}\right)^{1/2}$ | no waves | $2D/v^2$ |

Results in the above limits are corroborated by numerical simulation in *Figure 3C*, where we show initiation times for the limits above and for intermediate values of $vr_i/D$.

Lastly, we consider cells in 3D with 3D diffusion and find that there is a critical initial signaling colony size of $r_i = \sqrt{3}\,D/v$ below which the wave will not initiate. Around the critical colony size, $t_{\mathrm{init}}$ diverges as $(vr_i/D)^6[(vr_i/\sqrt{3}D)^2 - 1]^{-2}$. If $r_i \gg D/v$, then $t_{\mathrm{init}}$ again plateaus at a constant value $t_{\mathrm{min,3D}}$ that only depends on the characteristic time scale $D/v^2$:

$$t_{\mathrm{init}} \approx \begin{cases} \text{noinitiation} & r_i < \sqrt{3}\,D/v \\ \dfrac{(D/9\pi v^2)(vr_i/D)^6}{\left[(vr_i/\sqrt{3}D)^2 - 1\right]^2} & r_i \approx \sqrt{3}\,D/v \\ t_{\mathrm{min,3D}} = 2D/v^2 & r_i \gg \sqrt{3}\,D/v. \end{cases} \tag{12}$$

These analytic expressions again agree well with numerical simulation, as seen in *Figure 3C*. In Appendix 7: Initiation dynamics, we work out the case of cells in 2D with diffusion in 3D, for which there is a minimum initiating colony size of $r_i = D/v$. There, we also show that the qualitative findings presented above also hold for systems with discrete cells.

The critical initiating colony size for a 3D environment is reminiscent of elegant work on range expansions (*Tanaka et al., 2017*; *Barton and Turelli, 2011*). There, the effects of diffusive migration and population growth compete with each other, and a critical mass is needed to initiate the spatial advance of a particular genotype. Here, the dimension-dependent dynamics of concentration build-up dictate that a signaling wave which will always initiate in one- and two-dimensional environments requires a critical initial colony size in 3D.

Because the signaling wave always initiates in one- and two-dimensional environments, it can in principle be initiated by a single cell. As random activation of a single cell can initiate a signaling wave that fixes the entire population to maximal activation, these signaling dynamics have typically been thought of as unstable (*Deneke and Di Talia, 2018*). Yet, as we have shown here, even in one- and two-dimensional environments, the initiation time for colonies smaller than $D/v$ can be many orders of magnitude larger than the characteristic time scale of $D/v^2$ (*Figure 3D*). Thus, even though this signaling modality is technically unstable, it is robust against stochastic activation of a small number of cells over very long time scales.

In effect, then, even strictly positive-valued activation functions require a 'critical mass' of cells to initiate a signaling wave. In the context of neutrophil swarming – which we will shortly consider in more detail – this critical mass may provide a basis by which the immune system 'decides' whether to initiate a full-scale swarming response. In vitro experiments (*Reátegui et al., 2017*) indicate that small colonies of a pathogen can indeed fail to incite a swarm. Moreover, since the critical size of an initiating colony goes like $D/v$, we can see that relays utilizing smaller (lower molecular weight, higher $D$) signaling molecules require larger critical masses, all else equal.

Finally, we note that for cells with a Hill-like activation function $f(c) = c^n/(c^n + C_{\mathrm{th}}^n)$ of order $n \geq 2$, the above results for switch-like activation provide a good quantitative approximation of the initiation times (see Appendix 8: Wave initiation with Hill function activation). Moreover, for cells in 3D with Hill activation functions of order $n > 3$, there is a critical colony size just as for switch-like activation. These results highlight the role of spatial degrees of freedom in determining the wave initiation dynamics and stability.

## Application to neutrophil swarming and gradient generation

With a firm understanding of the diffusive wave and initiation dynamics, we now turn our sights to understanding a specific model system: neutrophil swarming. In beautiful work across several organisms (*Lämmermann et al., 2013*; *Isles et al., 2019*; *Reátegui et al., 2017*), experimentalists have observed striking behavior: an acute injury or infection can elicit rapid, highly directive motion of neutrophils – the most prevalent white blood cells – toward the site of the injury or infection. These experiments have demonstrated that a lipid small molecule called leukotriene B4 (LTB4) – along with many larger, slower-diffusing proteins (*Reátegui et al., 2017*) – governs the long-range recruitment of swarming neutrophils (*Lämmermann et al., 2013*; *Afonso et al., 2012*; *Reátegui et al., 2017*; *Isles et al., 2019*). Reategui et al. have noted the presence of several other pro- and anti-inflammatory lipid small molecules during swarming, though their precise roles are less clear. LTB4 serves to activate the neutrophils and also acts as a chemoattractant (*Afonso et al., 2012*) when receptors for

LTB4 are blocked, swarming behavior is significantly impaired (*Lämmermann et al., 2013*; *Reátegui et al., 2017*). The release of LTB4 has been thought to work as a relay, although the precise mechanistic details of this relay remain unclear (*Lämmermann and Germain, 2014*; *Kienle and Lämmermann, 2016*).

In vitro experiments performed with human neutrophils are particularly relevant given the results discussed so far. In these experiments, human-derived neutrophils are injected into a chamber, then settle onto the surface of a glass slide, resulting in a uniform sprinkling of cells in 2D. Also on the glass slide are circular 'targets' (of size $r_i$) coated in zymosan, a fungal surface protein that elicits a swarming response (*Reátegui et al., 2017*). Some cells land on or near the target, giving an initial condition as in *Figure 3A*. These cells begin signaling their neighbors, which in turn migrate towards the target (*Figure 4A*).

By tracking individual cells in time, one can deduce their migratory direction as a function of time. A typical metric for quantifying the directionality a cell's migration is the chemotactic index – the cosine of the angle $\theta$ between a cell's motion and the direction of the target (*Figure 4A*). One can average over the cells at a given distance $r$ and time $t$ to construct a plot of the average directionality $\langle \cos \theta \rangle$ in space and time. As pictured in *Figure 4B*, such a plot reveals a clear divide in space and time between cells that are highly directed toward the target (pink) and those without any particular directionality (white and light blue). We refer to the boundary of this divide as an information wave front – cells that lie underneath the curve have received the signal and begun chemotaxing toward the target while those above the curve have not.

Interestingly, the information wave front is convex with respect to the origin – a dramatic departure from what simple diffusive signaling by cells on the target would yield (*Figure 4A/B*), and from what Reategui et al. observe in experiments with neutrophils whose LTB4 receptors have been blocked (see Appendix 10: Simple diffusion model for more). We therefore posit that the cells may be participating in a relay in which they emit LTB4 in response to the same and check to see if this is consistent with the observed information wave front.

To do so, we perform a numerical simulation of *Equation (2)* with an additional term to account for the signaling of cells that land on the target. For this analysis, we assume a circular target of radius $r_i \approx 100$ µm, though the targets fabricated by Reategui et al. are smaller, oblong objects. Here, the diffusive environment is effectively three dimensional and the cells are close enough to allow for the use of a continuum model like *Equation (2)* (see below). Our model assumes switch-like activation of neutrophils, which we associate with the onset of directed chemotaxis. We ignore the inward migration of cells in this analysis, as it has a negligible effect on the information wave propagation since the cells move at a speed $u \approx 0.3$ µm/s $\ll v$ (see Appendix 11: Quantifying the effects of chemotaxis). Thus, as mentioned above, *Equation (2)* effectively has two parameters: $C_{\text{th}}/a\rho$ and $D$. Fitting these two parameters to the observed information wave front gives $C_{\text{th}}/a\rho \approx 3.67 \times 10^5$ s/m and $D \approx 1.25 \times 10^{-10}$ m²/s, the latter of which is consistent with the diffusion constant of a small molecule like LTB4. This implies a wave speed of $v \approx 1.7$ µm/s. Thus, we are validated in using a continuum model with a thick extracellular medium, as for this experiment the extracellular medium thickness $h = 2$ mm $\gg D/v$ and the mean distance between neutrophils, $d = 50$ µm, satisfies $vd/4D \approx 0.17 \ll 1$. The cell thickness $H \approx 10$ µm indeed satisfies $H \ll D/v$, meaning the use of the delta function to describe the cell distribution is valid. Finally, as LTB4 has a lifetime $1/\gamma$ of many minutes (*Bray, 1983*) and $D/v^2 \approx 40$ s $\ll 1/\gamma$, we can indeed ignore signaling molecule decay. These fit parameters give a curve that matches the transient dynamics over the field of view of the experiment (*Figure 4*). Thus, our relay model gives dynamics that are consistent with the dynamics of neutrophil swarming experiments – namely, the observed convex shape of the information wave front. Larger field-of-view and longer time-course experiments with varying cell densities and larger targets will provide a deeper mechanistic understanding of such relays, while also testing the scaling predictions of *Equation (5)* and *Equation (8)*.

The fit value of $C_{\text{th}}/a\rho = 3.67 \times 10^5$ s/m is consistent with the neutrophil's LTB4 receptor affinity. To show this, we first note that Reategui et al. measured the LTB4 emission rate under similar conditions as the relay experiment analyzed above; they found that $a \approx 40$ molecules per second per cell (see Appendix 9: Sensitivity of the information front to fit parameters for details). Using the cell density of $\rho = 1/d^2 = (50 \text{ µm})^{-2}$, we find that $C_{\text{th}} \approx 500$ pM. This value is within the range of the

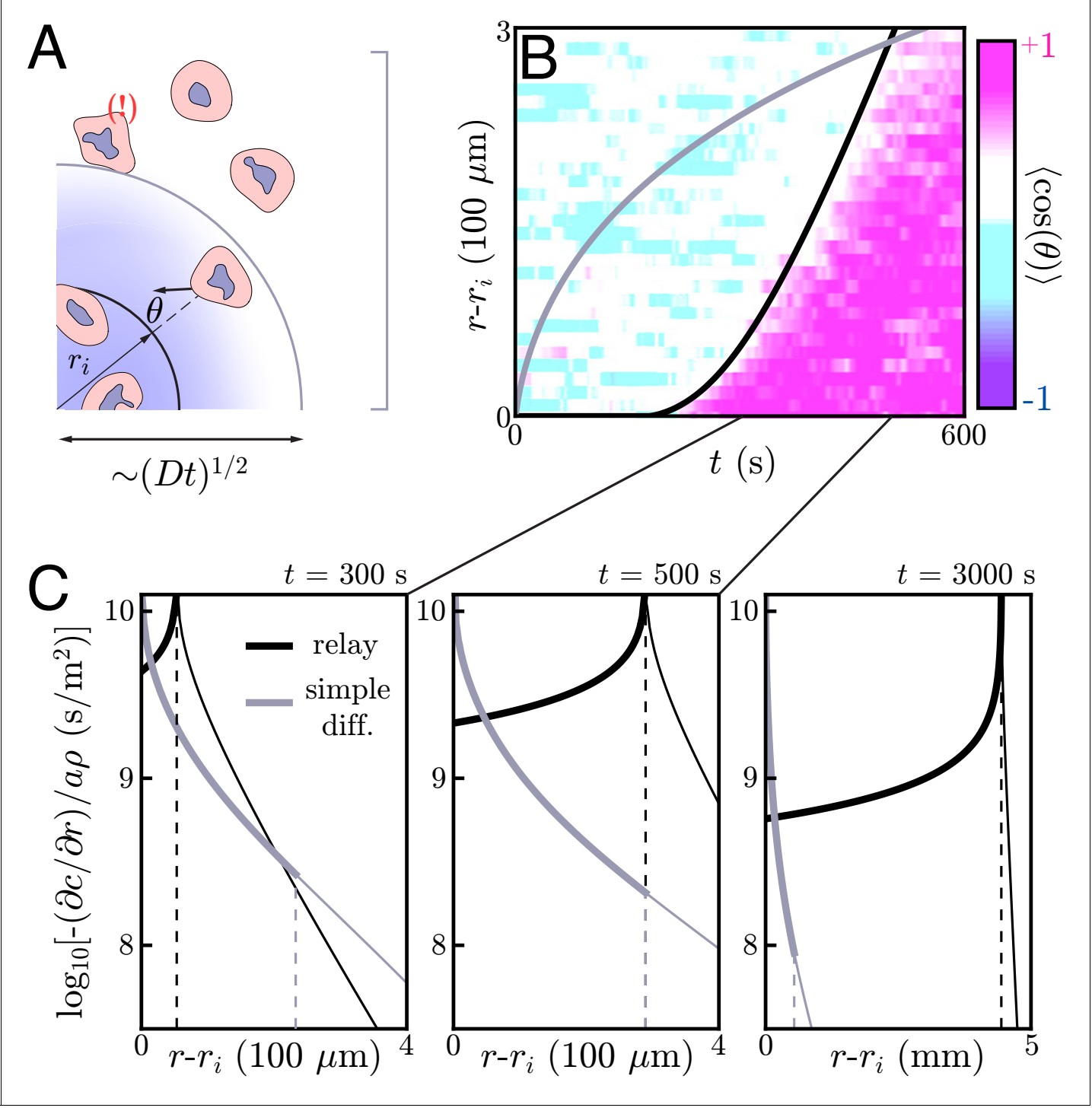

**Figure 4.** Application to neutrophil swarming. (**A**) Schematic of the simple diffusion model. Here, cells on the target (within $r_i$) signal distant neighbors by continuously emitting a single signaling molecule. If the neighboring cells have a chemotactic response, they migrate toward the target with some noise – that is, some non-zero angle $\theta$ with respect to the target. Otherwise, they move around with no sustained directionality. (**B**) Experimental data (color plot) reproduced from *Reátegui et al., 2017* showing the information wave front in neutrophil swarming experiments. By tracking the neutrophils in space and time, they observe highly directed motion of the neutrophils towards the target (pink) starting around $t = 200$ s. There is a clear boundary in space and time – the information wave front – between the regions where cells migrate toward the target (pink) and jostle around with no particular direction (white and light blue). While a relay theory (black line) is consistent with the convex shape of the information wave front, simple diffusive signaling by only the cells on the target (gray line) is not. The diffusion constants for both models is $D = 1.25 \times 10^{-10}$ m²/s. The threshold concentrations for the relay and simple diffusion models are $C_{\text{th}}/a\rho \approx 3.66 \times 10^5$ s/m and $2.91 \times 10^4$ s/m, respectively. The parameters for the relay

*Figure 4 continued on next page*

*Figure 4 continued*

model are chosen to fit the wave front by eye while the simple diffusion model parameters are chosen to give the same signaling distance at $t = 500$ s. (C) Gradients created by signaling relays (black) and simple diffusion (gray) models in panel B. The dashed vertical lines indicate the location of the information wave front. As time increases from left to right, the relay signaling motif gives an information wave that signals cells faster than simple diffusion in the long time limit. Cells within the wave front (to the left of the dashed lines that indicate the wave fronts) experience significantly larger gradients when the cells utilize a relay, which may facilitate efficient chemotaxis.

measured BLT1 receptor affinity for LTB4, which is reported to be approximately $0.1 - 2$ nM (*Yokomizo, 2015*).

Finally, we comment on the matter of *why* neutrophils might employ such signaling relays. As we have shown above, relays lead to 'fast' communication, in the sense that they give rise to diffusive waves which travel a distance $vt$ in a time $t$, compared to the $\sim \sqrt{Dt}$ distance of simple diffusion. However, there is another potential reason to use diffusive relays: they create strong gradients that may help cells chemotax effectively.

To get an idea of the gradients we are working with, we compare those generated by a relay – calculated by solving *Equation (2)* and approximated in *Equation (9)* – to a comparable simple diffusion model, such as that pictured in *Figure 4B*. (In Appendix 10: Simple diffusion model, we present the same comparison for a thin extracellular medium.) As is well-known, a burst-like emission of a diffusible molecule creates shallow, Gaussian concentration profiles away from the source; the same is true for continuous emission of a fixed source. Thus, the gradients that individual cells or small colonies of cells can create through simple diffusive signaling are orders of magnitude shallower than the collective gradients generated by relays (*Figure 4C*). This hints that neutrophils may use relays not solely for their improved signaling speed, but also for the strong resulting chemotactic gradients.

## Discussion

In this work, we have shown how simple cell signaling relays can give rise to diffusive waves whose properties are robust to many underlying details. Our work especially highlights the importance of the dimensionality of the extracellular medium, as seemingly innocent changes to the environment can have large effects on the resulting diffusive waves. The strong effect of system dimensionality is reminiscent of previous work on diffusive dynamics, which showed how dimensionality can effect Turing pattern instabilities (*Levine and Rappel, 2005*).

Although we have characterized the asymptotic dynamics, initiation, and potential design principles of these waves in several scenarios, many interesting problems remain as yet unsolved. First, as noted by Lammermann and colleagues (*Lämmermann and Germain, 2014*; *Kienle and Lämmermann, 2016*), it is unclear how the complexities of in vivo extracellular environments affect these results, particularly in the context of neutrophil swarming. Ambient flow (for example, in blood vessels), constrictions, and complex diffusive environments may lead to dynamics of biological relevance beyond those discussed here. Additionally, it would be interesting to study how different models of chemoreception and cellular uptake – topics of theoretical (*Muratov and Shvartsman, 2004*) and experimental (*Youk and Lim, 2014*; *Scherber et al., 2012*; *Tweedy et al., 2016*) relevance – affect our conclusions.

As an experimental test of our model, we propose studying neutrophil swarming dynamics over a wide field of view with varying cell densities and extracellular medium thicknesses. For diffusive waves with approximately our experimentally inferred parameters for neutrophil swarming ($D/v \approx 100$ µm), one could probe the thin extracellular medium limit of $h \ll D/v$ with microfluidic chambers of tens of microns in height. Similarly, with mm-scale chambers introduced by *Reátegui et al., 2017* and discussed in the previous section, one can reach the limit of a thick extracellular medium. Experiments in these two limits would provide quantitative tests of our theory. In particular, varying cell density would provide a test of the dimensionality-dependent relations for collective signaling wave speed, *Equations (5) and (8)*.

On a mechanistic level, although a relay mechanism would allow neutrophils to quickly coordinate their response, it remains unclear how inflammatory response is modulated in such a scenario. If inflammation during neutrophil swarming is governed by a fast-travelling wave, then how do the cells collectively turn off response? One possibility is that signaling pathways in neutrophil swarm

resolution – for instance, those involving LXA4 (*Reátegui et al., 2017*) production and emission – work by a similar relay mechanism; it is also possible that LTB4 production is governed by other fast-diffusing signaling molecules whose presence is necessary for LTB4 production, thereby limiting the relay's recruitment range.

Studies of the neutrophil relay mechanism may provide an interesting contrast to similar intercellular signaling dynamics in *Dictyostelium discoideum* (*Pálsson and Cox, 1996*; *Kessler and Levine, 1993*; *Noorbakhsh et al., 2015*) and microbial consortia (*Parkin and Murray, 2018*). The former provides a particularly striking contrast, since the waves that drive *Dictyostelium* signaling are pulsatile in nature, yet are also used to coordinate chemotactic response. Whereas continuous emission relays create continuous, steep concentration profiles, pulsatile relays in *Dictyostelium* create traveling wave packets of high concentration, each of which elicits a chemotactic response. We see no evidence of 'jumps' in chemotactic response during neutrophil swarming. It is not clear what drives one organism to adopt pulsatile signaling over relays with continuous emission, or vice versa.

Finally, it would also be interesting to leverage the design principles we have discussed for engineering synthetic relays, a field with a rich history (*Parkin and Murray, 2018*; *Brenner et al., 2008*; *Brenner et al., 2007*; *Basu et al., 2005*). To that end, our results provide a general framework for determining how system dimensionality, diffusion constants, activation functions, cell density, etc. affect cell signaling and wave initiation. Experimental work on this problem and others would provide tests of our many quantitative predictions.

## Materials and methods

To find the information wave front for cells in $n$ dimensions and diffusion in $m$ dimensions with continuous emission and Heaviside activation, we make use of the Green's function for the diffusion equation with sources in $n$ dimensions and diffusion in $m$ dimensions, $G_{n,m}(r,t;R,T)$. These equations are enumerated in Appendix 7: Initiation dynamics; $dT\,dR\,a\,\rho\,G_{n,m}(r,t;R,T)$ describes the concentration created at a radius $r$ and time $t$ by a tiny ring of sources at radius $R$ with density $\rho$ that emit at rate $a$ for duration $dT$ at time $T$.

To find the information front, one is looking for a curve $r_c(t)$ such that $C_{\text{th}} = c(r_c(t),t)$. Thus, with an initial signaling colony of size $r_i$, one must solve the problem:

$$C_{\text{th}} = a\rho \int_0^t dT \int_0^{\max[r_i, r_c(T)]} dR\, G_{n,m}(r_c(t),t;R,T). \tag{13}$$

This constraint equation considers every radius at time $T$ and, if it is less than $r_c(T)$, adds a concentration contribution of $a\rho\,dT\,dR\,G_{n,m}(r_c(t),t;R,T)$ at $r_c(t)$; the sum of all these contributions must be equal to $C_{\text{th}}$. If one wishes to find the information front for a simple diffusive theory, one performs the same integral as above, but truncates the integration over $R$ at $r_i$.

This method is preferable to brute PDE solving (for example, on a grid) since the former requires fine-grained meshing over the out-of-plane dimension when considering systems of, for example, cells in 2D and diffusion in 3D. In contrast, our Green's function method requires only numerical integration over the in-plane sources; the Green's functions appropriately keep track of the out-of-plane dynamics for us.

To solve this problem, we first find the initiation time, then find $r_c(t)$ at discrete times, incrementing in steps of $\Delta t \ll D/v^2$ (we use $\Delta t = D/10v^2$ in the main text and Appendices, which gives convergence of the information wave front). Linear interpolation between these points defines a continuous curve $r_c(t)$.

An explicit implementation of this method is provided at github./pdieterle/diffWavePropAndInit (*Dieterle, 2020*; copy archived at swh:1:rev:f8d9feffd57d05f47c8c14c6d9850643b2858d0a).

## Acknowledgements

We thank Richard Murray, James Parkin, Justin Bois, Mikhail Shapiro, David Nelson, Taekjip Ha, Johanna Dickmann, and Jenny Sheng for helpful discussions; Felice Frankel for help with figure preparation; and the reviewers of this manuscript for their thorough reading and helpful comments. We acknowledge support from the NSF through MRSEC DMR 14–20570 and the Kavli Foundation. PBD

is supported by the Paul M Young Fellowship through the Fannie and John Hertz Foundation. AA acknowledges support from NSF CAREER 1752024.

## Additional information

### Funding

| Funder | Grant reference number | Author |
| --- | --- | --- |
| National Science Foundation | MRSEC DMR 14-20570 | Ariel Amir |
| Kavli Foundation | | Ariel Amir |
| National Science Foundation | CAREER 1752024 | Ariel Amir |
| Fannie and John Hertz Foundation | Paul M. Young Fellowship | Paul B Dieterle |

The funders had no role in study design, data collection and interpretation, or the decision to submit the work for publication.

### Author contributions

Paul B Dieterle, Formal analysis, Investigation, Methodology, Writing - original draft, Writing - review and editing; Jiseon Min, Formal analysis, Investigation, Methodology, Writing - review and editing; Daniel Irimia, Conceptualization, Supervision, Methodology, Project administration, Writing - review and editing; Ariel Amir, Conceptualization, Formal analysis, Supervision, Funding acquisition, Investigation, Writing - original draft, Project administration, Writing - review and editing

### Author ORCIDs

Paul B Dieterle (iD) https://orcid.org/0000-0001-8129-7456
Daniel Irimia (iD) http://orcid.org/0000-0001-7347-2082
Ariel Amir (iD) https://orcid.org/0000-0003-2611-0139

### Decision letter and Author response

Decision letter https://doi.org/10.7554/eLife.61771.sa1
Author response https://doi.org/10.7554/eLife.61771.sa2

## Additional files

### Supplementary files

• Transparent reporting form

### Data availability

The only dataset we analyze or generate is present and available in Reategui (2017). PMID:29057147. Code for the figures is available at https://github.com/pdieterle/diffWavePropAndInit (copy archived at

https://archive.softwareheritage.org/swh:1:rev:f8d9feffd57d05f47c8c14c6d9850643b2858d0a/).

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

# Appendix 1

## Model set-up

Before doing any math, let's set up the scenarios we intend to study. We consider a continuum of cells described by a cell density, $\rho$. These cells emit one type of signaling molecule at a rate $a$ when the local concentration of the signaling molecule is above a certain threshold, $C_{\text{th}}$. The molecules diffuse in the extracellular medium with diffusion constant $D$. The concentration of the signaling molecule is described by the variable $c$, which is a function of both space and time: $c = c(\mathbf{r}, t)$. In general, then, we have

$$\frac{\partial c}{\partial t} = D\nabla^2 c + a\rho\Theta[c - C_{\text{th}}] \tag{14}$$

with $\Theta[.]$ the Heaviside step function. We study this model and variants going forward.

A word about notation before we proceed: in *Equation (14)*, we have written the source term as $a\rho\Theta[c - C_{\text{th}}]$, with $\rho$ the cell density. In some cases, we will write this source term slightly differently; for instance, with point-like cells homogeneously distributed in a two-dimensional plane, we will write the source term as $a\rho\,\delta(z)\Theta[c - C_{\text{th}}]$. Here, $\rho$ is a two-dimensional cell density, $\delta(.)$ is the Dirac delta function, and $z$ is the out-of-plane dimension. Therefore, the cell density $\rho$ which appears in all of the subsequent discussion will be a three-dimensional density if the cells are distributed in three dimensions, a two-dimensional density if the cells are distributed in two dimensions, and an one-dimensional density if the cells are distributed in one dimension. As we will show below, the choice of a delta function to describe the distribution in out-of-plane dimensions is justified when the cell size is small compared to the inherent length scale of the diffusive wave, $D/v$.

## Appendix 2

### Asymptotic wave ansatz

We start out by seeking to understand what dynamical properties a system described by *Equation (14)* has at large times. To study these dynamics, we need an inspired guess for what the dynamics will look like. We imagine that when a small volume of cells starts signaling its neighbors – and those neighbors start signaling their neighbors – that a reasonable guess for the dynamics of such a signaling relay is an outward propagating wave with speed *v*. We define *r* as the distance from the center of the outward propagating wave. At long times for a uniform cell density, the shape information wave front will obey radial symmetry and our ansatz becomes $c(\mathbf{r}, t) = c(\mathbf{r} - vt\hat{r})$ with $\hat{r}$ the unit vector pointing from the origin to the wave front.

With this guess, we can define a new coordinate $\tilde{r} = r - vt$ (we call this $\tilde{x} = x - vt$ for cells in one dimension) which defines the distance to the wave front. Note that $\tilde{r} < 0$ means we are inside the wave front while $\tilde{r} > 0$ means we are beyond it. With these definitions, $\partial c/\partial \tilde{r} = \partial c/\partial r$ and $\partial c/\partial t = -v\,\partial c/\partial \tilde{r}$. For cells in 1D, we consider the *y* and *z* to be dimensions perpendicular to the line of cells with the density described by $\rho\,\delta(y)\delta(z)$ with $\rho$ measured in cells per unit length; for cells in 2D, we consider *z* to be the out-of-plane dimension and the density to be described by $\rho\delta(z)$ with $\rho$ measured in cells per unit area; for cells in 3D, $\rho$ is measured in cells per unit volume. Assuming azimuthal symmetry in 2D and radial symmetry in 3D, we arrive at

$$\text{cells in 1D} : 0 = D\left(\frac{\partial^2 c}{\partial \tilde{x}^2} + \frac{\partial^2 c}{\partial y^2} + \frac{\partial^2 c}{\partial z^2}\right) + v\frac{\partial c}{\partial \tilde{x}} + a\rho\,\delta(y)\delta(z)\Theta[c - C_{\text{th}}] \tag{15a}$$

$$\text{cells in 2D} : 0 = D\left(\frac{\partial^2 c}{\partial \tilde{r}^2} + \frac{1}{r}\frac{\partial c}{\partial \tilde{r}} + \frac{\partial^2 c}{\partial z^2}\right) + v\frac{\partial c}{\partial \tilde{r}} + a\rho\,\delta(z)\Theta[c - C_{\text{th}}] \tag{15b}$$

$$\text{cells in 3D} : 0 = D\left(\frac{\partial^2 c}{\partial \tilde{r}^2} + \frac{2}{r}\frac{\partial c}{\partial \tilde{r}}\right) + v\frac{\partial c}{\partial \tilde{r}} + a\rho\,\Theta[c - C_{\text{th}}] \tag{15c}$$

These equations can be simplified once more by noting that we are considering asymptotic – that is, large *r* – dynamics. Thus, as long as $v \gg D/r$, we can say that $v\partial c/\partial \tilde{r}$ dominates terms like $D(\partial c/\partial \tilde{r})/r$ and we can ignore the latter (*Tanaka et al., 2017*). By construction, our ansatz says that $c(\tilde{r} = 0) \equiv C_{\text{th}}$, meaning that $\Theta[c - C_{\text{th}}] = \Theta[-\tilde{r}]$. This gives simplified equations according to:

$$\text{cells in 1D} : 0 = D\left(\frac{\partial^2 c}{\partial \tilde{x}^2} + \frac{\partial^2 c}{\partial y^2} + \frac{\partial^2 c}{\partial z^2}\right) + v\frac{\partial c}{\partial \tilde{x}} + a\rho\,\delta(y)\delta(z)\Theta[-\tilde{x}] \tag{16a}$$

$$\text{cells in 2D} : 0 = D\left(\frac{\partial^2 c}{\partial \tilde{r}^2} + \frac{\partial^2 c}{\partial z^2}\right) + v\frac{\partial c}{\partial \tilde{r}} + a\rho\,\delta(z)\Theta[-\tilde{r}] \tag{16b}$$

$$\text{cells in 3D} : 0 = D\frac{\partial^2 c}{\partial \tilde{r}^2} + v\frac{\partial c}{\partial \tilde{r}} + a\rho\,\Theta[-\tilde{r}]. \tag{16c}$$

These equations provide both a natural length scale, $D/v$, and a natural timescale, $D/v^2$. One can see that these are the relevant time and length scales for our problem by non-dimensionalizing, for example, *Equation (16c)* to get

$$\text{cells in 3D} : 0 = \frac{\partial^2 (cv^2/a\rho D)}{\partial (v\tilde{r}/D)^2} + \frac{\partial (cv^2/a\rho D)}{\partial (v\tilde{r}/D)} + \Theta[-v\tilde{r}/D]. \tag{17}$$

Thus, every length scale in the problem is normalized by $D/v$; because of our traveling wave ansatz, every length scale can be converted to a time scale by dividing by *v*, giving $D/v^2$ as the natural timescale. The natural length scale is useful, for example, for understanding what it means for cells to be in 'one dimension' or for diffusion to be in 'two dimensions'. If the cells are organized in a line (or on a plane) such that their average deviation from the line (or distance from the plane) is

$d \ll D/v$, then they are effectively in one (or two) dimensions and *Equation (16a)* (or *Equation (16b)*) holds. If cells are constricted to a narrow channel of width $h \ll D/v$ (or an extracellular medium of thickness $h \ll D/v$), then diffusion is effectively one (or two) dimensional. For instance, for cells confined in a very narrow one-dimensional channel of width $h \ll D/v$, we can simplify *Equation (16a)* because $\partial^2 c / \partial y^2 = \partial^2 c / \partial z^2 = 0$ and $\delta(y)\delta(z) \to 1/h^2$. The resulting equation is the exact same as for cells in 3D but with a source term proportional to $a\rho/h^2$:

$$\text{cells in 1D, diffusion in 1D} : 0 = D\frac{\partial^2 c}{\partial \tilde{x}^2} + v\frac{\partial c}{\partial \tilde{x}} + \frac{a\rho}{h^2}\Theta[-\tilde{x}]. \tag{18}$$

For cells in 2D with an extracellular medium of thickness $h \ll D/v$, diffusion is effectively two-dimension and, by the same logic that produced *Equation (18)*, the asymptotic governing equation is:

$$\text{cells in 2D, diffusion in 2D} : 0 = D\frac{\partial^2 c}{\partial \tilde{r}^2} + v\frac{\partial c}{\partial \tilde{r}} + \frac{a\rho}{h}\Theta[-\tilde{r}]. \tag{19}$$

As *Equations (17), (18), and (19)* are all the same, the dynamics of cells in 1D with diffusion in 1D are the same as those of cells in 2D with diffusion in 2D or those in 3D with diffusion in 3D.

Moreover, the non-dimensional *Equation (17)* shows us that the concentration scale of interest is $a\rho D/v^2$, meaning that there must be a relationship along the lines of $C_{\text{th}} \sim a\rho D/v^2 v \sim (a\rho D/C_{\text{th}})^{1/2}$ – exactly the wave speed relationship we found in the main text.

One can, however, arrive at a different governing equation by considering cells in 2D (for example, cells sitting on a plane) with a thick extracellular medium of thickness $h \gg D/v$. In this case, diffusion effectively takes place in three dimensions and

$$\text{cells in 2D, diffusion in 3D} : 0 = D\left(\frac{\partial^2 c}{\partial \tilde{r}^2} + \frac{\partial^2 c}{\partial z^2}\right) + v\frac{\partial c}{\partial \tilde{r}} + a\rho\delta(z)\Theta[-\tilde{r}]. \tag{20}$$

which, by the same logic above, is functionally equivalent to the governing equation for cells in 1D with diffusion in 2D. We can therefore see that it is not the dimensionality of the cell distribution or the diffusive environment that determines the asymptotic dynamics, but rather the difference in dimension between the two.

Going forward, we will think of cells in two dimensions, as we have done in the main text. This will allow us to interpolate between an effectively two-dimensional diffusive environment (the thin extracellular medium limit) and an effectively three-dimensional environment (the thick extracellular medium limit).

## Cells in 2D, diffusion in 2D: the thin extracellular medium limit

For an extracellular medium of thickness $h \ll D/v$, diffusion effectively takes place in two dimensions as argued in the previous section. The signaling molecule concentration has no $z$-dependence and concentrations get normalized by $h$. Here,

$$0 = D\frac{\partial^2 c}{\partial \tilde{r}^2} + v\frac{\partial c}{\partial \tilde{r}} + \frac{a\rho}{h}\Theta[-\tilde{r}] \tag{21}$$

is our asymptotic governing equation.

For both $\tilde{r}<0$ and $\tilde{r}>0$, *Equation (21)* reduces to two straightforward-to-solve linear ODEs. With $b_i$ as constants that we will determine momentarily,

$$\tilde{r}<0 : \ 0 = D\frac{\partial^2 c}{\partial \tilde{r}^2} + v\frac{\partial c}{\partial \tilde{r}} + a\rho/h \implies c(\tilde{r}<0) = b_2 e^{-v\tilde{r}/D} + b_3 - a\rho\tilde{r}/hv \tag{22a}$$

$$\tilde{r}>0 : \ 0 = D\frac{\partial^2 c}{\partial \tilde{r}^2} + v\frac{\partial c}{\partial \tilde{r}} \implies c(\tilde{r}>0) = b_0 e^{-v\tilde{r}/D} + b_1. \tag{22b}$$

We can solve for the $b_i$ by applying boundary conditions. First, we demand $c \to 0$ as $\tilde{r} \to \infty$ and that the concentration profile only blow up linearly as $\tilde{r} \to -\infty$. Physically, these demands are justified

as follows: the concentration as $\tilde{r} \to \infty$ has to go to zero because there are no cells emitting in that region and it is far from the wave front; the concentration as $\tilde{r} \to -\infty$ can grow at most linearly because the cells a distance $-\tilde{r}$ from the wave front have only been emitting for a time $-\tilde{r}/v$. Combining these asymptotic boundary conditions with the demand that $c(\tilde{r})$ be continuous and have a continuous first derivative at $\tilde{r} = 0$ allows us to stitch together the solutions in *Equations (22a)* and *Equations (22b)* to yield:

$$c(\tilde{r} < 0) = a\rho D/hv^2 - a\rho \tilde{r}/hv \tag{23a}$$

$$c(\tilde{r} > 0) = a\rho D e^{-v\tilde{r}/D}/hv^2. \tag{23b}$$

We show this concentration profile in the left panel of *Appendix 3—figure 1A*. From the above, we infer that

$$C_{\mathrm{th}} = c(0) = a\rho D/hv^2 \implies v = \sqrt{a\rho D/hC_{\mathrm{th}}}. \tag{24}$$

Thus, we have an explicit formula relating wave speed, emission rate, cell density, diffusion constant, extracellular medium thickness, and threshold concentration. We can also see that the concentration profile beyond the wave front is exponential, not Gaussian as for simple diffusion. The concentration inside the wave front grows linearly as the distance from the wave front.

## Cells in 2D, diffusion in 3D: the thick extracellular medium limit

Next, we consider *Equation (16b)* in the limit that the extracellular medium $h \gg D/v$. In this limit, we effectively have cells in 2D with diffusion in 3D. With cells sitting on a substrate, signaling molecules can only diffusive in the upper half of the plane, and we have a semi-infinite environment which accounts for an extra factor of 2 in the emission term, yielding:

$$0 = D\left(\frac{\partial^2 c}{\partial \tilde{r}^2} + \frac{\partial^2 c}{\partial z^2}\right) + v\frac{\partial c}{\partial \tilde{r}} + 2a\rho\,\delta(z)\Theta[-\tilde{r}]. \tag{25}$$

Instead of working directly with $\delta(z)$, we consider the cells to be of a thickness $H$ such that $2\delta(z) \to \frac{1}{H}\sqrt{\frac{2}{\pi}}\exp(-z^2/2H^2)$ and

$$0 = D\left(\frac{\partial^2 c}{\partial \tilde{r}^2} + \frac{\partial^2 c}{\partial z^2}\right) + v\frac{\partial c}{\partial \tilde{r}} + \frac{a\rho}{H}\sqrt{\frac{2}{\pi}}e^{-z^2/2H^2}\Theta[-\tilde{r}] \tag{26}$$

Next, we take a partial Fourier transform of *Equation (26)* with $k$ and $C(\tilde{r}, k)$ the Fourier partners of $z$ and $c(\tilde{r}, z)$, respectively. Here, we choose $C(\tilde{r}, k) \equiv \frac{1}{\sqrt{2\pi}}\int_{-\infty}^{\infty} e^{ikz}c(\tilde{r}, z)$. This gives:

$$0 = D\left(\frac{\partial^2 C}{\partial \tilde{r}^2} - k^2 C\right) + v\frac{\partial C}{\partial \tilde{r}} + \sqrt{\frac{2}{\pi}}e^{-H^2k^2/2}a\rho\,\Theta[-\tilde{r}] \tag{27}$$

which is another pair of piecewise, straightforward-to-solve, linear ODEs. Solving with the same boundary conditions that yielded *Equations (23a)* and *Equations (23b)*, we arrive at

$$C(\tilde{r} < 0, k) = \frac{a\rho e^{-H^2k^2/2}}{Dk^2\sqrt{2\pi}}\left(2 - \frac{\left(1 + \sqrt{4D^2k^2/v^2 + 1}\right)\exp\left(\frac{v\tilde{r}}{2D}\left(\sqrt{4D^2k^2/v^2 + 1} - 1\right)\right)}{\sqrt{4D^2k^2/v^2 + 1}}\right) \tag{28a}$$

$$C(\tilde{r} > 0, k) = \frac{a\rho e^{-H^2k^2/2}}{Dk^2\sqrt{2\pi}}\frac{\sqrt{4D^2k^2/v^2 + 1} - 1}{\sqrt{4D^2k^2/v^2 + 1}}\exp\left(-\frac{v\tilde{r}}{2D}\left(\sqrt{4D^2k^2/v^2 + 1} + 1\right)\right). \tag{28b}$$

To find the concentrations at the cells, we can take the inverse partial Fourier transform of these expressions at $z = 0$. But first, we note that the right sides of *Equations (28a)* and *Equations (28b)* have no support when $k \gg v/D$. Thus, if $Hv/D \ll 1$, the term $e^{-H^2k^2/2}$ is irrelevant for calculating the

real-space concentrations and can be replaced with 1. This is equivalent to having chosen $\delta(z)$ to describe the out-of-plane cell density.

Proceeding with $e^{-H^2k^2/2} \to 1$, one can take the inverse partial Fourier transform and arrive at

$$C_{\text{th}} = 2a\rho/\pi v. \tag{29}$$

Similarly, in the limit $|\tilde{r}| \gg D/v$,

$$c(\tilde{r} \ll -D/v, z) \approx \frac{2a\rho}{v}\sqrt{\frac{-\tilde{r}v}{\pi D}}\left(e^{vz^2/4D\tilde{r}} - \sqrt{-\frac{\pi vz^2}{4D\tilde{r}}}\text{erfc}\sqrt{-\frac{vz^2}{4D\tilde{r}}}\right) \tag{30a}$$

$$c(\tilde{r} \gg D/v, z) \approx a\rho\sqrt{\frac{D}{\pi\tilde{r}v^3}}e^{-v\tilde{r}/D}e^{-vz^2/4D\tilde{r}}. \tag{30b}$$

The former has the same functional dependence on $z$ as the concentration a distance $z$ away from a continuously emitting point source with diffusion in 1D after a time $\tilde{r}/v$ (see Appendix 6: Assessing the validity of a continuum analysis). Using the above, we can find the concentration in the plane of the cells ($z = 0$):

$$c(\tilde{r} \ll D/v, z = 0) \approx \frac{2a\rho}{v}\sqrt{\frac{-\tilde{r}v}{\pi D}} \tag{31a}$$

$$c(\tilde{r} \gg D/v, z = 0) \approx a\rho\sqrt{\frac{D}{\pi\tilde{r}v^3}}e^{-v\tilde{r}/D}. \tag{31b}$$

We show this concentration profile in the left panel of *Appendix 3—figure 1B*.

Of course, one can arrive at the scaling form of *Equation (29)* through dimensional analysis considerations, as discussed in the main text. Equivalently, one can non-dimensionalize *Equation (25)* along the lines of *Equation (17)*, as we will now. By normalizing all length scales by $D/v$ and all concentration scales by $a\rho/v$, one can show that *Equation (25)* is equivalent to

$$0 = \left(\frac{\partial^2(cv/a\rho)}{\partial(\tilde{r}v/D)^2} + \frac{\partial^2(cv/a\rho)}{\partial(zv/D)^2}\right) + \frac{\partial(cv/a\rho)}{\partial(\tilde{r}v/D)} + 2\delta(zv/D)\Theta[-v\tilde{r}/D]. \tag{32}$$

from which we can tell that the concentration scales of the problem, including $C_{th}$, are proportional to $a\rho/v$, and thus that $v \sim a\rho/C_{\text{th}}$, exactly as required by *Equation (29)*.

## Cells in 1D, diffusion in 3D: an artificial case

Finally, we consider a line of cells in one dimension with diffusion taking place in three dimensions. This corresponds to a somewhat artificial test case of cells in a line with mean distance from the line $d \ll D/v$ and diffusion in an environment of size $h \gg D/v$ in the dimensions perpendicular to this line of cells. Nonetheless, it is interesting because we have to include the finite size of the cells in order to get a traveling wave solution. Here,

$$0 = D\left(\frac{\partial^2 c}{\partial \tilde{x}^2} + \frac{\partial^2 c}{\partial y^2} + \frac{\partial^2 c}{\partial z^2}\right) + v\frac{\partial c}{\partial \tilde{x}} + \frac{a\rho}{\pi H^2}e^{-(y^2+z^2)/2H^2}\Theta[-\tilde{r}] \tag{33}$$

with $v$ the wave speed; $\tilde{x}$ the distance to the wave front; $y$ and $z$ the extra diffusive dimensions; and $H$ the size of the cells.

Taking partial Fourier transforms across both $y$ and $z$ gives with the Fourier transform conventions and notation used above gives

$$0 = D\left(\frac{\partial^2 C}{\partial \tilde{x}^2} - k_y^2 C - k_z^2 C\right) + v\frac{\partial C}{\partial \tilde{x}} + \frac{a\rho}{\pi}e^{-(k_y^2+k_z^2)H^2/2}\Theta[-\tilde{x}] \tag{34}$$

which reduces to *Equation (27)* with $k^2 \to k_y^2 + k_z^2$. One can then find the concentration profiles and

self-consistency relationship for $C_{\text{th}}$ by inverse Fourier transforming the analogs of *Equations (31a)* and *Equations (31b)*. The value of $C_{\text{th}} = c(\tilde{x} = 0, y = z = 0)$ diverges as $H \to 0$.

## Appendix 3

### Pulsed emission and decay

In this section, we consider pulsed emission and decay of the signaling molecule. These scenarios are relevant for signaling pathways in, for example, *Dictyostelium* (*Pálsson and Cox, 1996*; *Noorbakhsh et al., 2015*; *Kessler and Levine, 1993*) and *E. coli* (*Parkin and Murray, 2018*), in which intracellular dynamics produce a pulse-like release of signaling molecules into the extracellular medium.*Kessler and Levine, 1993* have previously used this machinery to construct a signaling model for *Dictyostelium*, including pulsed emission and signaling molecule decay. Here, we consider the effects of each independently.

We explicitly discuss only the asymptotics of cells in 2D with diffusion in 2D (equivalent to cells in 1D with diffusion in 1D or cells in 3D with diffusion in 3D, as shown previously), although we quote the results for cells in 2D with diffusion in 3D (equivalent to cells in 1D with diffusion in 2D) which are obtained using the Fourier transform machinery in Appendix 2: Asymptotic wave ansatz. Here again, the asymptotic dynamics depend on the difference in dimensionality between the cellular and the diffusive environment.

### Pulsed emission with cells in 2D and diffusion in 2D

Here, we consider a square pulse of length $\tau$ emitted once a cell exceeds the threshold concentration $C_{\text{th}}$. In the moving frame, this pulse has length $v\tau$ – for notational simplicity here, we dispense with dimensional subscripts on the wave speed – giving rise to the pulsed emission analog of *Equation (21)*:

$$0 = D\frac{\partial^2 c}{\partial \tilde{r}^2} + v\frac{\partial c}{\partial \tilde{r}} + \frac{a\rho}{h}\Theta[-\tilde{r}]\Theta[\tilde{r}+v\tau] \tag{35}$$

which is nothing more than three piecewise linear equations, which we stitch together as before. The source term is zero when $\tilde{r}<-v\tau$ (Region I) or $\tilde{r}>0$ (Region III) and $a\rho_0$ for $-v\tau<\tilde{r}<0$ (Region II). We thus recover the following:

$$\text{Region I}: c(\tilde{r}<-v\tau) = b_1 + b_2 e^{-v\tilde{r}/D} \tag{36a}$$

$$\text{Region II}: c(-v\tau<\tilde{r}<0) = b_3 + b_4 e^{-v\tilde{r}/D} - a\rho\tilde{r}/hv \tag{36b}$$

$$\text{Region III}: c(\tilde{r}>0) = b_5 + b_6 e^{-v\tilde{r}/D} \tag{36c}$$

Applying the same boundary conditions as with continuous emission, we arrive at

$$\text{Region I}: c(\tilde{r}<-v\tau) = a\rho\tau/h \tag{37a}$$

$$\text{Region II}: c(-v\tau<\tilde{r}<0) = a\rho D/hv^2 - \frac{a\rho D}{hv^2}e^{-v^2\tau/D}e^{-v\tilde{r}/D} - a\rho\tilde{r}/hv \tag{37b}$$

$$\text{Region III}: c(\tilde{r}>0) = \frac{a\rho D}{hv^2}(1-e^{-v^2\tau/D})e^{-v\tilde{r}/D} \tag{37c}$$

which tells us a few things of interest. First – as seen in the right panel of *Appendix 3—figure 1 A* (here, $\tau = 2D/v^2 = 2C_{\text{th}}/a\rho$) – the concentration profile for $\tilde{r}<-v\tau$ is flat. (As with continuous emission, pulsed emission gives the familiar exponential profile beyond the wave front.) Second, the wave speed, pulse width, cell density, emission rate, extracellular medium thickness, and threshold concentration are related through the equation

$$C_{\text{th}} = a\rho D(1 - e^{-v^2\tau/D})/hv^2. \tag{38}$$

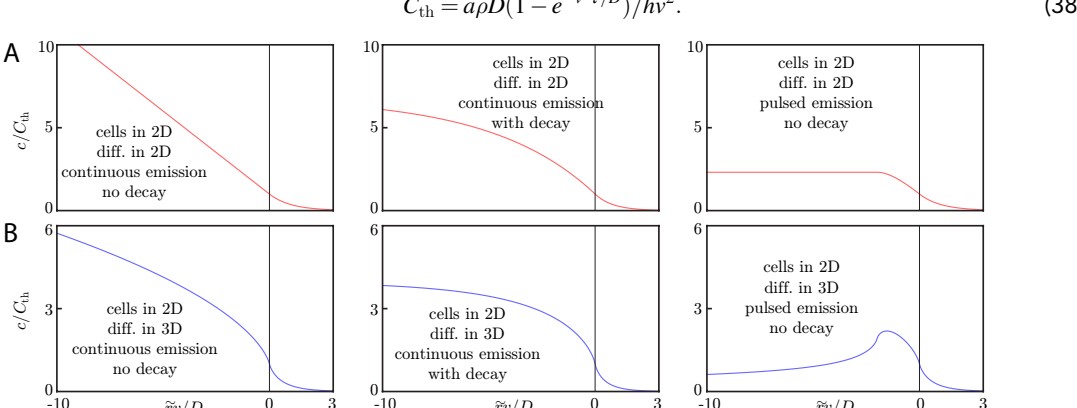

**Appendix 3—figure 1.** Relay concentration profiles. (**A**) Signaling molecule concentration profiles for cells in 2D and diffusion in 2D. Here, we assume continuous emission and no signaling molecule decay (left panel), continuous emission and signaling molecule decay (middle panel), or pulsed emission with no signaling molecule decay (right panel). The decay constant for the middle panel is $\gamma = v^2/4D$ while the pulse width is $\tau = 2D/v$ in the right panel. For the left, center, and right panels, the threshold concentration is calculated according to *Equations (24), (41), and (38)*, respectively. As discussed in the main text, the concentration profiles flatten (with respect to the profile generated by continuous emission without decay) inside the wave front once decay or pulsed emission is accounted for. (**B**) Signaling molecule concentration profiles for the same cases as in A, but with diffusion in 3D. Compared to the case of continuous emission without decay, the concentration profiles flatten (when accounting for decay) or have a local maximum (in the case of pulsed emission).

For $\tau \gg D/v^2$, we recover the usual relationship of $C_{\text{th}} = a\rho D/hv^2$. In region 2, the profile will grow linearly as before until $-v\tilde{r}/D$ becomes comparable to $v^2\tau/D$, at which point $e^{-v^2\tau/D - v\tilde{r}/D}$ becomes of order unity and the profile levels off.

To understand how the wave speed with pulsed emission, $v$, compares to the wave speed with continuous emission, $(a\rho D/hC_{\text{th}})^{1/2}$, we have plotted $v/(a\rho D/hC_{\text{th}})^{1/2}$ as a function of $1/\tau$ in *Appendix 3—figure 2A*. We have normalized $\tau$ by a characteristic time $\tau_c = hC_{\text{th}}/a\rho$, which is equal to $D/v^2$ for continuous emission. When $\tau < \tau_c$, the wave speed goes to zero. There is no wave-like solution for shorter pulses.

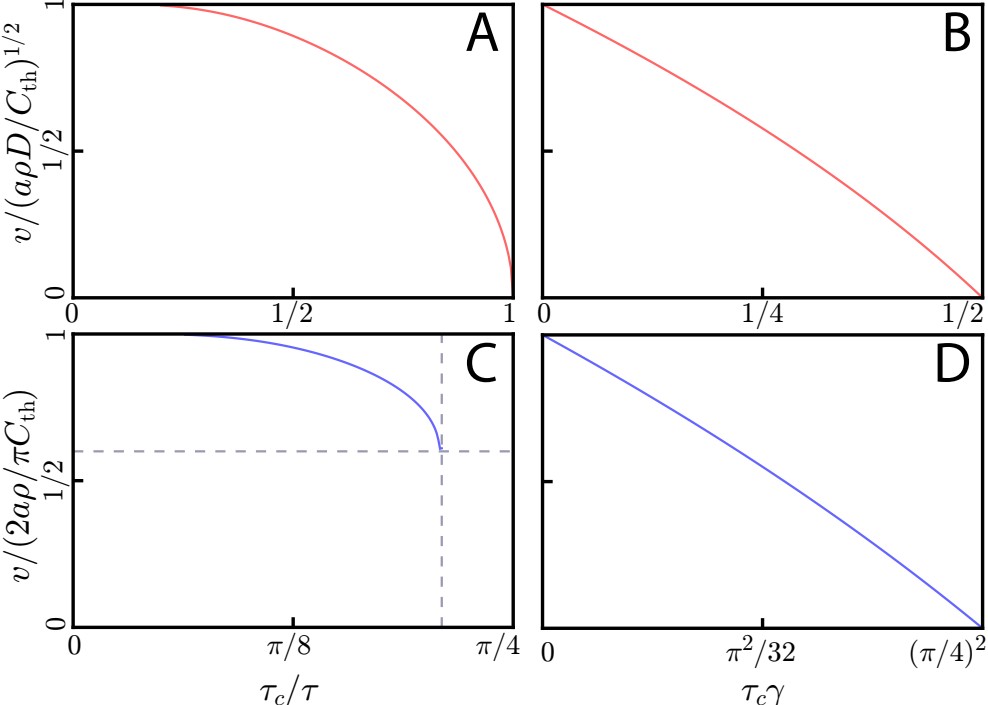

**Appendix 3—figure 2.** Wave speed with pulsed emission or decay. (**A**) Wave speed $v$ for square pulse emission by cells in 2D with diffusion in 2D as a function of pulse width $\tau$. We normalize $v$ by the $\tau \to \infty$ wave speed of $(a\rho D/C_{\text{th}})^{1/2}$. At $\tau = \tau_c = hC_{\text{th}}/a\rho$, the wave speed goes to zero, and for shorter pulses there is no wave-like solution. (**B**) Wave speed for continuous emission by cells in 2D with diffusion in 2D, accounting for signaling molecule decay at rate $\gamma$. For $\gamma = t_{\text{min,2D}} = 1/2\tau_c$, with $\tau_c$ as in panel A, the wave speed goes to zero and there are no wave-like solutions for larger decay rates. (**C**) Wave speed $v$ as a function of pulse width $\tau$ for square pulse emission of a signaling molecule by cells in 2D with a 3D diffusive environment. At $\tau \approx 1.53\tau_c = 1.53D(2C_{\text{th}}/\pi a\rho)^2$ (vertical dashed line), there is a minimum wave speed of $v \approx 0.6 \times (2a\rho/\pi C_{\text{th}})$ (horizontal dashed line), unlike with 2D diffusion. Importantly, $\tau \approx 1.53\tau_c$ is longer than the minimum initiation time of $4\tau_c/\pi$. Thus, for values of $\tau$ between these two, cells can reach the threshold concentration but cannot propagate a traveling information wave. (**D**) Wave speed $v$ as a function of decay rate $\gamma$ for cells in 2D with diffusion in 3D. At $\gamma = (\pi/4)^2/\tau_c$, with $\tau_c$ as in C, the wave speed goes to zero.

We note that a timed pulsed emission considered here is formally equivalent to cells signaling until the local concentration exceeds $c(\tilde{r} = -v\tau) = a\rho\tau/h$. This is relevant in, for example, quorum sensing models in which the local presence of a signaling molecule can both upregulate (at relatively low concentrations) and downregulate (at relatively high concentrations) release of the same signaling molecule (*Parkin and Murray, 2018*).

## Continuous emission plus decay with cells in 2D and diffusion in 2D

At last, we characterize the effect of signaling molecule decay at rate $\gamma$ by adding a term of $-\gamma c$ to *Equation (16b)*. In the thin extracellular medium limit,

$$0 = D\frac{\partial^2 c}{\partial \tilde{r}^2} + v\frac{\partial c}{\partial \tilde{r}} + \frac{a\rho}{h}\Theta[-\tilde{r}] - \gamma c. \tag{39}$$

This is another piecewise set of linear differential equations, which we can solve as without decay to yield:

$$c(\tilde{r}<0) = \frac{a\rho}{h\gamma} - \frac{a\rho}{2h\gamma}\left[1 + (1 + 4D\gamma/v^2)^{-1/2}\right]e^{\frac{\tilde{r}v}{2D}\left(\sqrt{4D\gamma/v^2+1}-1\right)} \tag{40a}$$

$$c(\tilde{r}>0) = \frac{a\rho}{2h\gamma}\left[1 - (1+4D\gamma/v^2)^{-1/2}\right]e^{-\frac{\tilde{r}v}{2D}\left(\sqrt{4D\gamma/v^2+1}+1\right)} \tag{40b}$$

as the concentration profiles and

$$C_{\text{th}} = \frac{a\rho}{2h\gamma}\left[1 - (1+4D\gamma/v^2)^{-1/2}\right]v = 4D\gamma\left[\left(1 - \frac{2h\gamma C_{\text{th}}}{a\rho}\right)^{-2} - 1\right]^{-1} \tag{41}$$

as our wave speed relationship. The concentration profile is flatter than its decay-free counterpart (*Appendix 3—figure 1A*). For $\gamma \ll v^2/D$, *Equation (41)* gives the decay-free relationship. And – as with pulsed emission – the wave speed goes to zero, this time when $\gamma \to 1/2\tau_c$ where $\tau_c = hC_{\text{th}}/a\rho$ (*Appendix 3—figure 2B*).

## Pulsed emission with cells in 2D and diffusion in 2D

Here,

$$0 = D\left(\frac{\partial^2 C}{\partial \tilde{r}^2} - k^2 C\right) + v\frac{\partial C}{\partial \tilde{r}} + \sqrt{2/\pi}\, a\rho\, \Theta[-\tilde{r}]\Theta[v\tau + \tilde{r}] \tag{42}$$

which we can solve to yield:

$$\text{Region I}: C(\tilde{r} < -v\tau, k) = B_1(k)e^{\frac{\tilde{r}v}{2D}\left(-1+\sqrt{1+4D^2k^2/v^2}\right)} \tag{43a}$$

$$\text{Region II}: C(-v\tau < \tilde{r} < 0, k) = \sqrt{\frac{2}{\pi}}\frac{a\rho_0}{Dk^2} + B_2(k)e^{-\frac{\tilde{r}v}{2D}\left(1+\sqrt{1+4D^2k^2/v^2}\right)}$$
$$+ B_3(k)e^{\frac{\tilde{r}v}{2D}\left(-1+\sqrt{1+4D^2k^2/v^2}\right)} \tag{43b}$$

$$\text{Region III}: C(\tilde{r} > 0, k) = B_4(k)e^{-\frac{\tilde{r}v}{2D}\left(1+\sqrt{1+4D^2k^2/v^2}\right)} \tag{43c}$$

with $B_i(k)$ chosen such that $C$ and its first derivative are continuous:

$$B_1(k) = \frac{a\rho}{D\sqrt{2\pi}}\frac{\left(v+\sqrt{4D^2k^2+v^2}\right)\left(e^{\frac{v^2\tau}{2D}\left(\sqrt{4D^2k^2/v^2+1}-1\right)}-1\right)}{k^2\sqrt{4D^2k^2+v^2}} \tag{44a}$$

$$B_2(k) = \frac{a\rho}{D\sqrt{2\pi}}\frac{\left(v-\sqrt{4D^2k^2+v^2}\right)e^{-\frac{v^2\tau}{2D}\left(\sqrt{4D^2k^2/v^2+1}+1\right)}}{k^2\sqrt{4D^2k^2+v^2}} \tag{44b}$$

$$B_3(k) = -\frac{a\rho}{D\sqrt{2\pi}}\frac{v+\sqrt{4D^2k^2+v^2}}{k^2\sqrt{4D^2k^2+v^2}} \tag{44c}$$

$$B_4(k) = \frac{a\rho}{D\sqrt{2\pi}}\frac{\left(v-\sqrt{4D^2k^2+v^2}\right)\left(1-e^{-\frac{v^2\tau}{2D}\left(\sqrt{4D^2k^2/v^2+1}-1\right)}\right)}{k^2\sqrt{4D^2k^2+v^2}} \tag{44d}$$

Inverse Fourier transforming at $z=0$ gives the real-space concentration and the following self-consistency relationship for the wave speed:

$$C_{\text{th}} = \frac{1}{\sqrt{2\pi}}\int_{-\infty}^{\infty} dk\, B_4(k), \tag{45}$$

which simplifies to
$C_{\text{th}} = 2a\rho/\pi v$ for $\tau \gg D/v^2$.

We can compare $\tau$ to the characteristic time $\tau_c = D(\pi C_{th}/2a\rho)^2$, which is equal to $D/v^2$ in the limit of continuous emission. Numerical solution of *Equation (45)* reveals that there is no self-consistent solution to *Equation (45)* until about $\tau \approx 1.53\tau_c$, at which point $v \approx 0.6 \times 2a\rho/\pi C_{th}$; $\tau \approx 1.53\tau_c$ is larger than the minimum initiation time of $t_{min,3D} = 4\tau_c/\pi$ (*Appendix 3—figure 2 C*, Section Appendix 7: Initiation dynamics). Thus, an initial pulse of length $4\tau_c/\pi < \tau < 1.53\tau_c$ from cells within an initial signaling radius $r_i$ can cause neighboring cells to exceed $C_{th}$, but cannot trigger a wave-like solution asymptotically.

## Continuous emission plus decay with cells in 2D and diffusion in 3D

We can add signaling molecule decay to the embedded system dynamics by adding a term of $-\gamma C(\tilde{r}, k)$ to *Equation (27)* with $\gamma$ the signaling molecule decay rate. Going through the same exercise yields:

$$C(\tilde{r}<0,k) = \sqrt{\frac{2}{\pi}\frac{a\rho}{\gamma + k^2 D}} - \frac{a\rho}{\sqrt{2\pi}}\frac{\sqrt{4D^2k^2/v^2 + 4D\gamma/v^2 + 1} + 1}{(Dk^2 + \gamma)\sqrt{4D^2k^2/v^2 + 4D\gamma/v^2 + 1}}e^{\frac{\tilde{r}v}{2D}\left(-1 + \sqrt{4D^2k^2/v^2 + 4D\gamma/v^2 + 1}\right)} \tag{46a}$$

$$C(\tilde{r}>0,k) = \frac{a\rho}{\sqrt{2\pi}}\frac{\sqrt{4D^2k^2/v^2 + 4D\gamma/v^2 + 1} - 1}{(Dk^2 + \gamma)\sqrt{4D^2k^2/v^2 + 4D\gamma/v^2 + 1}}e^{-\frac{\tilde{r}v}{2D}\left(1 + \sqrt{4D^2k^2/v^2 + 4D\gamma/v^2 + 1}\right)} \tag{46b}$$

with

$$C_{th} = \frac{a\rho}{\pi\sqrt{D\gamma}}\arcsin\left[\left(1 + v^2/4D\gamma\right)^{-1/2}\right] \tag{47}$$

as the parameter relationship obtained after an inverse Fourier transform at $z=0$ and $x=0$. This gives a profile that propagates as a pulse (*Appendix 3—figure 1B*).

In the limit $\gamma \ll v^2/D$, we recover the familiar expression $C_{th} = 2a\rho/\pi v$. Again, as when we accounted for signaling molecule decay with cells in 2D and diffusion in 2D, the wave speed approaches zero, but with $\gamma = (\pi/4)^2/\tau_c$ where $\tau_c = D(\pi C_{th}/2a\rho)^2$ (*Appendix 3—figure 2 D*).

## Appendix 4

### Finite extracellular medium

We consider now what happens when one does not lie in the extreme cases of an extracellular medium of thickness $h \gg D/v$ or $h \ll D/v$. To examine the case of arbitrary thickness $h$, we turn to the method of images (*Appendix 4—figure 1A*).

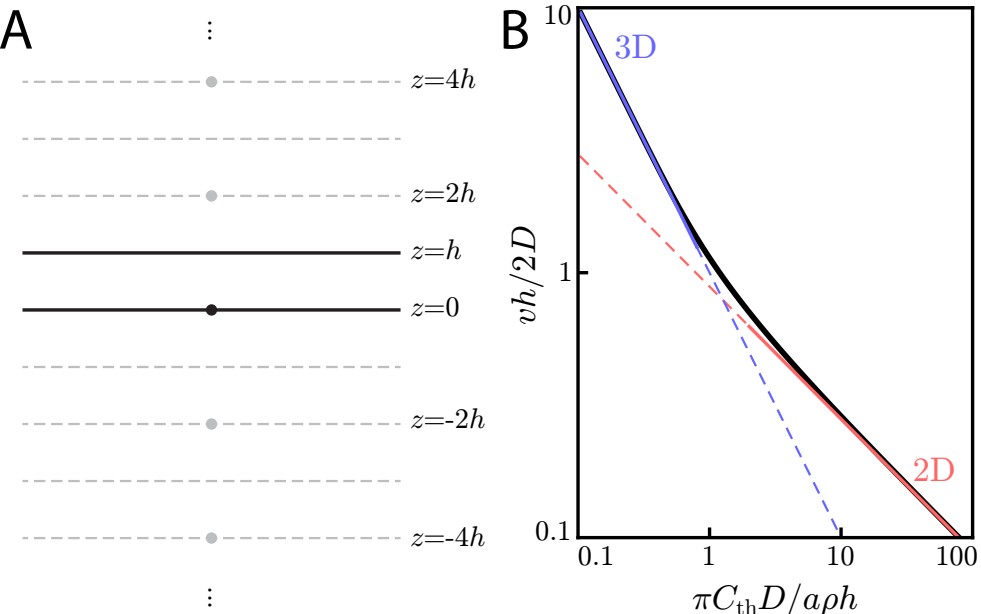

**Appendix 4—figure 1.** Wave speed in a finite extracellular medium. (**A**) Method of images configuration for solving for concentrations in a finite-sized extracellular medium of thickness $h$. A point-like source (a cell) at $z = 0$ is placed in an extracellular medium of thickness $h$. In order to calculate the concentration with the appropriate boundary conditions of $\partial c/\partial z$ at $z = 0, h$, one need only add the contributions from 'image cells' at $z = \pm 2jh$ for $j = 1, 2, \dots$. (**B**) Universal curve (black line) showing non-dimensionalized wave speed ($vh/2D$) versus non-dimensionalized threshold concentration ($\pi C_{\text{th}} D/a\rho h$). In the limit of $vh/2D \gg 1$, we recover the familiar 3D scaling law of *Equation (29)* (blue line). In the limit of $vh/2D \ll 1$, we get the 2D scaling law of *Equation (24)* (red line).

Our boundary conditions for the extracellular medium require the concentration to obey $\partial c/\partial z = 0$ at $z = 0, h$ – signaling molecules cannot diffuse through the boundaries. By invoking the uniqueness theorem, we know that if we can find an arrangement of 'image cells' – each emitting diffusible signaling molecule at rate $a$ – that satisfies these boundary conditions, then this arrangement of cells gives the *unique* solution for the concentration profile inside the extracellular medium. In our case, to satisfy the boundary condition above, we have image cells at $z = \pm 2jh$ for $j = 1, 2, \dots$.

This means that we can find the concentration profiles simply by adding up the contributions from many discrete sources. Given this knowledge, we seek a relationship like *Equation (24)* or *Equation (29)* but for arbitrary $h$. To do so, we use Green's function integration and the fact that we can analyze the asymptotic dynamics of cells in 1D to deduce the asymptotic dynamics of cells in 2D, as previously shown.

The concentration – as measured at $(r, z = 0, t)$ – of a burst-like emission by a single point-like source at $(R, 2jh, T)$ is given by the Green's function:

$$G(r, z = 0, t; R, 2jh, T) = \frac{e^{-\frac{(r-R)^2 + (2jh)^2}{4D(t-T)}}}{2\pi D(t-T)}, \tag{48}$$

so the Green's function of a single point-like source in a finite-thickness extracellular medium is (*Appendix 4—figure 1A*):

$$G_h(r,z=0,t;R,T) = \sum_{j=-\infty}^{\infty} G(r,z=0,t;R,2jh,T) = \frac{e^{-\frac{(r-R)^2}{4D(t-T)}}}{2\pi D(t-T)} \left( 1 + 2\sum_{j=1}^{\infty} e^{\frac{-(jh)^2}{D(t-T)}} \right). \tag{49}$$

We assume a traveling wave solution at speed $v$ meaning that the concentration $C_{\text{th}} = c(r=vt,z=0,t\to\infty)$ is, for a density of cells $\rho$ emitting with rate $a$, given by:

$$C_{\text{th}} = c(vt,0,t\to\infty) = a\rho \lim_{t\to\infty} \int_0^{vt} dR \int_{R/v}^t dT \, G_h(vt,0,t;R,T) =$$

$$\frac{a\rho}{2\pi D} \int_{-\infty}^0 d\tilde{R} \int_0^{-\tilde{R}/v} \frac{d\tilde{t}}{\tilde{t}} e^{-\frac{\tilde{R}^2}{4D\tilde{t}}} \left( 1 + 2\sum_{j=1}^{\infty} e^{-\frac{(jh)^2}{D\tilde{t}}} \right) \tag{50}$$

with the substitutions $\tilde{t} = t - T$ and $\tilde{R} = R - vt$. This yields:

$$C_{\text{th}} = \frac{2a\rho}{\pi v}\left( 1 + 2\sum_{j=1}^{\infty} \int_0^{-\infty} dx \, \text{Ei}\left[\frac{1}{x}\left(\frac{jhv}{2D}\right)^2 + x\right] \right) = \frac{2a\rho}{\pi v}\sum_{j=-\infty}^{\infty} \int_0^{-\infty} dx \, \text{Ei}\left[\frac{1}{x}\left(\frac{jhv}{2D}\right)^2 + x\right] \tag{51}$$

for $x = v\tilde{R}/4D$ and $\text{Ei}[.]$ the exponential integral function. For $h \gg D/v$, *Equation (51)* reduces to *Equation (29)* because the term $\sum_{j=1}^{\infty} \cdots \approx 0$. Meanwhile, for $h \ll D/v$, the sum over $j$ can be turned into an integral, giving the familiar thin extracellular medium relationship, *Equation (24)*.

We emphasize that *Equation (51)* provides a universal relationship between threshold concentration, wave speed, cell density, and signaling molecule emission rate for any extracellular medium thickness. By dividing both sides of *Equation (51)* by $a\rho h/\pi D$, we arrive at a relationship between a non-dimensionalized threshold concentration, $\pi C_{\text{th}} D/a\rho h$, and a non-dimensionalized wave speed, $vh/2D$:

$$\frac{\pi C_{\text{th}} D}{a\rho h} = \frac{2D}{vh}\sum_{j=-\infty}^{\infty} \int_0^{-\infty} dx \, \text{Ei}\left[\frac{1}{x}\left(\frac{jhv}{2D}\right)^2 + x\right]. \tag{52}$$

We plot this relationship in *Appendix 4—figure 1B* and see that *Equation (52)* is an interpolation between the thin ($h \ll D/v$) and thick ($h \gg D/v$) extracellular medium limits.

# Appendix 5

## Asymptotic wave dynamics with Hill function activation
### Numerical solutions show traveling waves

As shown in the main text, making the change from a Heaviside function source term to an order-$n$ Hill function ($\Theta[c - C_{\text{th}}] \to c^n/(c^n + C_{\text{th}}^n)$) in *Equation (14)* preserves the scaling relationships *Equations (24) and (29)* with a constant factor as long as the new source terms give traveling wave solutions. We have found numerically that $n \geq 1$ Hill functions indeed give traveling wave solutions, with $n = 1, 2, 3$ shown for thin and thick extracellular media in *Appendix 5—figure 1*.

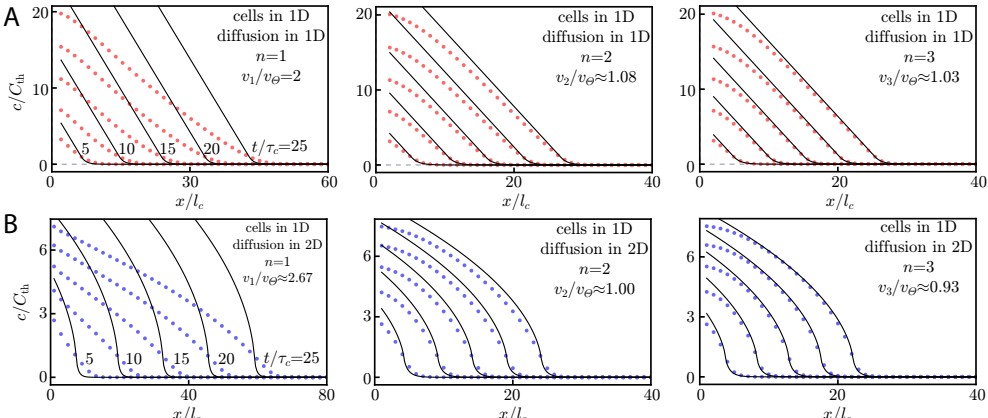

**Appendix 5—figure 1.** Wave speeds and profiles with Hill function activation. (**A**) Numerical simulation of cells in 1D with diffusion in 1D and Hill function activation. The details of the simulation are described in Appendix 5: Asymptotic wave dynamics with Hill function activation. For $n \geq 2$, we see good agreement between the Heaviside theory (black lines) and the Hill function numerics (red dots). Snapshots are shown at $t/\tau_c = 5, 10, 15, 20, 25$ where $\tau_c = h^2 C_{\text{th}}/a\rho$ equals $D/v_\Theta^2$, the characteristic time scale for Heaviside activation. Note that the x-axis is scaled by the characteristic length $l_c = h(DC_{\text{th}}/a\rho)^{1/2}$, which is the length scale for Heaviside activation with a delta function source, $D/v_\Theta$. In the insets, we display the wave speed for the order-$n$ Hill function, $v_n$, compared to the Heaviside activation wave speed, $v_\Theta = (a\rho D/h^2 C_{\text{th}})^{1/2}$. Our fit to the $n = 1$ data gives $v_{n=1} \approx 1.93 v_\Theta$, but we have shown in Appendix 5: Asymptotic wave dynamics with Hill function activation that $v_{n=1} = 2v_\Theta$, meaning that the wave speeds in the insets are slight underestimates. (**B**) Numerical simulation of cells in 1D and diffusion in 2D with Hill function activation. For $n \geq 2$, the wave speed and concentration profiles (blue dots) agree well with the Heaviside theory (black lines). The theory plotted here assumes a delta-function-like source with respect to the extra diffusive dimension, as in *Equation (25)*. The numerics, however, use a very narrow ($H = l_c/10$) Gaussian source. Here, the characteristic length $l_c = \pi h D C_{\text{th}}/2a\rho$ equals $D/v_\Theta$, the length scale for Heaviside activation. The x-axis is scaled by the same quantity. Snapshots are shown at $t/\tau_c = 5, 10, 15, 20, 25$ with the characteristic time $\tau_c = D(\pi h C_{\text{th}}/2a\rho)^2$ equal to $D/v_\Theta^2$, the time scale for Heaviside activation. In the insets, we display the wave speed for the order-$n$ Hill function, $v_n$, compared to the Heaviside activation wave speed, $v_\Theta = 2a\rho/\pi C_{\text{th}}$.

To find these solutions, we numerically solved *Equation (14)* with a Hill function source term for cells in 1D with diffusion in one or two dimensions. We used $D = 10^{-10}$ m$^2$/s and $v_\Theta = 2$ $\mu$m/s with the threshold concentration determined by *Equations (29) and (24)* with $v \to v_\Theta$. In this way, we could compare $v_n$ – the wave speed given by the order-$n$ Hill function – to the Heaviside wave speed $v_\Theta$. For our numerics, we imposed a maximum size step of $D/10v_\Theta$ for the spatial dimension where the cells live, a maximum step size of $D/30v_\Theta$ for the added spatial dimension (when modeling diffusion in 2D), and a maximum time step of $D/10v_\Theta^2$. These step sizes give convergence of the information wave fronts, which we define as the curves $r_c(t)$ such that $c(r_c(t), t) = C_{\text{th}}$. We simulated times $t_{\max} \leq 25D/v_\Theta^2$ using Mathematica's 'NDSolveValue' function and, when modeling diffusion in 2D, replaced the delta function in *Equation (14)* with a Gaussian of width $D/10v_\Theta$. As noted earlier, since

the width of this Gaussian is sufficiently smaller than D/v, the numerical results will be close to the delta function limit; indeed, this substitution gives a wave speed (according to an inverse Fourier transform of *Equation (28b)* at $\tilde{r}, z = 0$) of $v/v_\Theta \approx 0.95$, in close correspondence with the Delta function wave speed. The initiating colony is of size $r_i = 2D/v_\Theta$, and we assume all cells in the initiating colony signal at the maximal rate $a$.

To find the wave speeds noted in *Equation (1)*, we found the location of the wave front, $r_c(t)$, and fit a line to the region of $24D/v_\Theta^2 \leq t \leq 25D/v_\Theta^2$. As shown in *Appendix 5—figure 1*, for $n \geq 2$, the wave speeds are very close to $v_\Theta$ with significant deviation only for $n = 1$. Even the concentration profiles are in good agreement with the Heaviside solution for $n \geq 2$.

## Connection to Fisher waves

When the dimensionality of the cell distribution matches the diffusive dimensionality and cell activation is described by the $n = 1$ Hill function, one can find the wave speed by using a modified version of the analysis pioneered by *Fisher, 1937* and *Kolmogorov et al., 1937*. To see this, we first consider a modified version of *Equation (16c)* with order-$n$ Hill function activation:

$$0 = D\frac{\partial^2 c}{\partial \tilde{r}^2} + v\frac{\partial c}{\partial \tilde{r}} + a\rho\frac{c^n}{c^n + C_{\text{th}}^n}. \tag{53}$$

The new activation term is mathematically obnoxious as it no longer has a simple spatial interpretation; with a Heaviside function, for example, one can turn a term like $\Theta[c - C_{\text{th}}]$ into a simple function of $\tilde{r}$: $\Theta[c - C_{\text{th}}] = \Theta[-\tilde{r}]$. This has an important consequence: instead of solving two differential equations with constant source terms and matching boundary conditions (as we did for the Heaviside emission), we must now solve a single differential equation with a difficult non-linear source term. We note that *Equation (53)* is, in the limit $n \to \infty$, equivalent to a relay with Heaviside activation.

However, there is a distinct advantage to the new source term: it is one-to-one in $c$. Thus, we may make the substitution $f = \frac{c^n}{c^n + C_{\text{th}}^n}$, $c = C_{\text{th}}\left(\frac{f}{1-f}\right)^{1/n}$, then plug this into *Equation (53)* to yield (after some rearrangement):

$$0 = v\frac{\partial f}{\partial \tilde{r}} + D\left[\frac{\partial^2 f}{\partial \tilde{r}^2} + \frac{1 + (2f-1)n}{nf(1-f)}\left(\frac{\partial f}{\partial \tilde{r}}\right)^2\right] + \frac{na\rho}{C_{\text{th}}}f^{2-1/n}(1-f)^{1+1/n}. \tag{54}$$

*Equation (54)* looks a lot like the traditional Fisher equation,

$$0 = v\frac{\partial f}{\partial \tilde{r}} + D\frac{\partial^2 f}{\partial \tilde{r}^2} + \frac{na\rho}{C_{\text{th}}}f(1-f) \tag{55}$$

in that it has a source term that goes to zero at $f \to \{0,1\}$, a term $D\frac{\partial^2 f}{\partial \tilde{r}^2}$, and a term $v\frac{\partial f}{\partial \tilde{r}}$. We therefore take Fisher's approach (*Fisher, 1937*) and think of gradients of $f$ as functions of $f$ rather than $x$. As such, we define $F(f) = \frac{\partial f}{\partial x}$, which allows us to make the substitution $\frac{\partial^2 f}{\partial x^2} = \frac{\partial F}{\partial x} = \frac{\partial f}{\partial x}\frac{\partial F}{\partial f} = F\frac{\partial F}{\partial f}$. This is valid under the assumption that the concentration profiles are monotone decreasing, which is seen to be the case in *Appendix 5—figure 1*. Under all of the above, we get:

$$0 = vF + D\left[F\frac{\partial F}{\partial f} + \frac{1 + (2f-1)n}{nf(1-f)}F^2\right] + \frac{na\rho}{C_{\text{th}}}f^{2-1/n}(1-f)^{1+1/n} \tag{56}$$

which gives us a non-linear ODE for $F$. Of particular note are the boundary conditions for $F$. Namely, $F(0) = F(1) = 0$, which is to say that cells well inside of the wave front are all emitting at their maximal rate since $c \gg C_{\text{th}}$ and that cells well beyond the wave front are not emitting at all. Thus, there is no spatial dependence on the cellular activation, $f = c^n/(c^n + C_{\text{th}}^n)$, in these regions.

Next, we turn to the traditional method of examining the $f \to 0$ limit. As $F \to 0$, *Equation (56)* becomes, to lowest order in $f$ and assuming $F \approx -\lambda f^\beta$,

$$0 = -\lambda v + D\left[\lambda^2 \beta f^{\beta-1} + \frac{1-n}{n}\lambda^2 f^{\beta-1}\right] + \frac{na\rho}{C_{\text{th}}}f^{2-1/n-\beta}. \tag{57}$$

where $\lambda > 0$. One can only obtain a self-consistency relationship between $v$ and $C_{\text{th}}/a\rho$ if $\beta = 2 - 1/n$. Otherwise, $f^{2-1/n-\beta}$ diverges or goes to zero. With this choice, *Equation (57)* becomes:

$$n = 1 : 0 = -\lambda v + D\lambda^2/n + \frac{a\rho}{C_{\text{th}}}\lambda = \frac{1}{2D}\left(v \pm \sqrt{v^2 - \frac{4a\rho D}{C_{\text{th}}}}\right) \tag{58a}$$

$$n > 1 : 0 = -\lambda v + \frac{na\rho_0}{C_{\text{th}}} + \mathcal{O}(f^{1-1/n})\lambda = \frac{na\rho}{vC_{\text{th}}} \tag{58b}$$

where in *Equation (58b)* we have taken the $f \to 0$ limit. To get a wave speed, $v$, out of *Equation (58a)*, we demand that the quantity under the square root be non-negative, which ensures that $\lambda > 0$ is a real number as assumed. This means $v \geq 2\sqrt{a\rho D/C_{\text{th}}} = 2v_\Theta$ – a bound that is very similar to conventional Fisher waves. In the case of Fisher waves, the minimum wave speed is selected for (*Fisher, 1937*; *Kolmogorov et al., 1937*); the same is true here, as the minimum wave speed $v = 2\sqrt{a\rho D/C_{\text{th}}}$ is what one finds after numerically solving the 1D dynamics (*Appendix 5—figure 1A*). In contrast, for $n > 1$, this method yields no such wave speed bound.

# Appendix 6

## Assessing the validity of a continuum analysis

Next, we consider the validity of a continuum analysis like *Equation (16b)* for studying asymptotic wave dynamics. To do so, we compare our continuum wave speed relationships, *Equations (24) and (29)*, to a simple model of discrete cells on a lattice in 1D (*Appendix 6—figure 1A*). We refer to this as the discrete lattice model, which has been studied previously and is the subject of on-going work (*Keener, 2000*; Dieterle P and Amir A, 2020. Manuscript in preparation). We briefly discuss the results of this lattice model below, and show that it agrees with the continuous model when the separation between cells, *d*, is much less than the characteristic length $D/v$. This heuristic also holds for cells in two and three dimensions, and for cells scattered randomly according to a Poisson process.

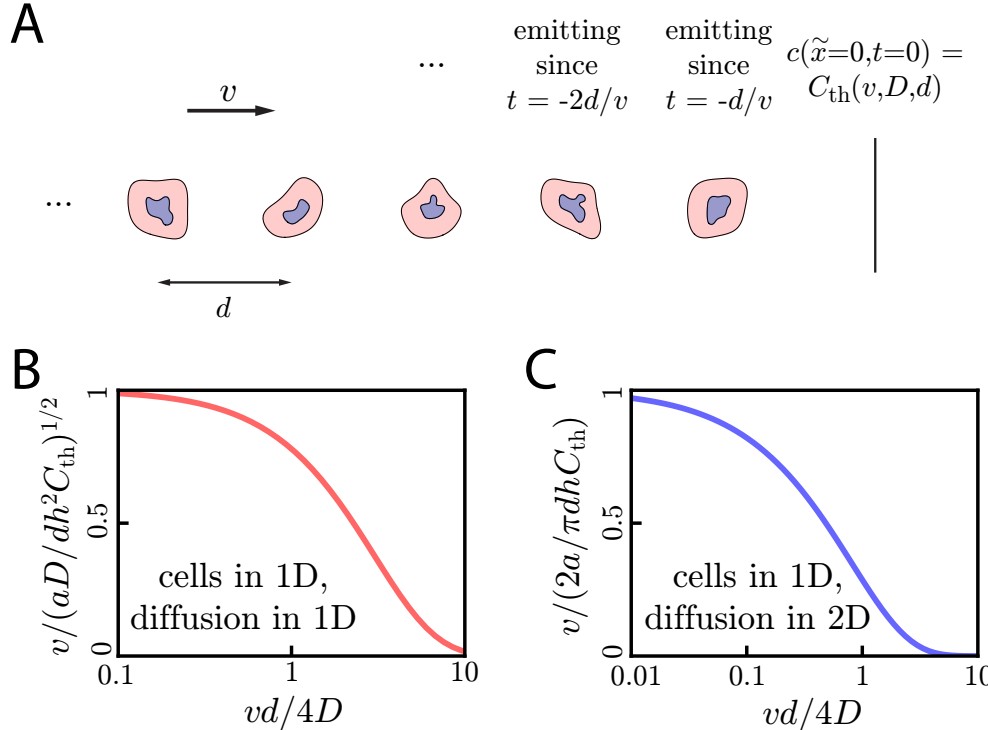

**Appendix 6—figure 1.** Wave dynamics with discrete sources. (**A**) Cartoon showing the discrete lattice model of cell signaling relays for cells in 1D. Here, a concentration wave propagates rightward at speed *v* through a group of cells spaced by a constant distance *d*. The concentration at $\tilde{x} = t = 0$ defines the threshold concentration in this discrete system and is a function of $v$, $D$, and $d$: $C_{\text{th}} = C_{\text{th}}(v, D, d)$. (**B**) Wave speed *v* compared to the continuous theory value of $(aD/dh^2 C_{\text{th}})^{1/2}$ as a function of $vd/4D$ for cells in 1D with diffusion in 1D. We can see that the wave speed approaches the continuous theory value for $vd/4D \ll 1$. (**C**) Same as B but for cells in 1D and diffusion in 2D.

To start, we first calculate the three-dimensional concentration (SI units of $1/\text{m}^3$) generated by a continuously emitting point source emitting at a rate *a* at a distance *x* after a time *t*. For diffusion in *m* dimensions, we will refer to this concentration as $c_{\bullet,m}(x,t)$. For diffusion in $m = 2$ dimensions, we will consider a semi-infinite environment in order to recapitulate *Equation (29)*, which holds for cells in one (two) dimensions with diffusion in a semi-infinite two-dimensional (three-dimensional) space. These relationships are:

$$c_{\bullet,1}(x,t) = \frac{a}{h^2} \int_0^t dT \, \frac{e^{-x^2/4DT}}{(4)^{1/2}} = \frac{a}{h^2} \sqrt{\frac{t}{\pi D}} \left( e^{-x^2/4Dt} - \sqrt{\frac{\pi x^2}{4Dt}} \text{erfc} \sqrt{\frac{x^2}{4Dt}} \right) \tag{59a}$$

$$c_{\bullet,2}(x,t) = \frac{2a}{h}\int_0^t dT\, \frac{e^{-x^2/4DT}}{4\pi DT} = -\frac{a}{2\pi hD}\mathrm{Ei}\left(-\frac{x^2}{4Dt}\right) \tag{59b}$$

with $\mathrm{erfc}[.]$ the complementary error function and $\mathrm{Ei}[.]$ the exponential integral. We have assumed one-dimensional diffusion takes place in a narrow channel with cross-sectional area $h^2$ and two-dimensional diffusion takes place in an extracellular medium of thickness $h$.

Next, we assume the cells in this lattice model perform a signaling relay with Heaviside activation: once the local concentration exceeds a threshold $C_{\mathrm{th}}$, they participate in the signaling molecule emission at a rate $a$. If the resulting wave speed is $v$, then a cell at a distance $\tilde{x} = -jd$ from the wave front has been emitting for a time $jd/v$. That cell then creates a concentration $c_{j,m} = c_{\bullet,m}(jd, jd/v)$ at $(\tilde{x}, t) = (0, 0)$. The full concentration at the wave front is $C_{\mathrm{th}}$ by definition, and it is equal to the sum of the concentrations created by all cells behind the wave front. This gives us the following self-consistency relationships between the threshold concentration, $C_{\mathrm{th}}$; wave speed, $v$; diffusion constant, $D$; and cell separation $d$:

$$\mathrm{diff.in1D}: C_{\mathrm{th}} = \sum_{j=1}^{\infty} c_{j,1} = \frac{a}{vh^2}\sum_{j=1}^{\infty}\sqrt{j\frac{vd}{\pi D}}\left[e^{-jvd/4D} - \sqrt{j\frac{vd}{4D}}\,\mathrm{erfc}\left(\sqrt{j\frac{vd}{4D}}\right)\right] \tag{60a}$$

$$\mathrm{diff.in2D}: C_{\mathrm{th}} = \sum_{j=1}^{\infty} c_{j,2} = -\frac{a}{2}\sum_{j=1}^{\infty}\mathrm{Ei}\left(-j\frac{vd}{4D}\right). \tag{60b}$$

*Equations (60a)* and (60b) provide relationships analogous to *Equations (24) and (29)*. In fact, in the limit $vd/4D \ll 1$, the sums in these relationships are well-approximated by an integral over $j$ from $j = 0$ to $\infty$. In this limit, with $\rho = 1/d$, *Equation (60a)* becomes $C_{\mathrm{th}} = a\rho D/h^2 v^2$, the one-dimensional analog of *Equation (24)*; similarly, *Equation (60b)* simplifies to $C_{\mathrm{th}} = 2a\rho/\pi hv$, the one-dimensional analog of *Equation (29)*. (One can turn *Equation (60b)* into an integral from $j = 0$ to $\infty$. The integrand diverges at $j = 0$, but is still integrable because the divergence is logarithmic.) Therefore, we can see that the continuum limit is valid when the separation between cells is $d \ll 4D/v$. We have shown the approach to the continuous theory limit in *Appendix 6—figure 1B/C*.

## Appendix 7

### Initiation dynamics

In this section, we demonstrate the initiation time relationships discussed in the main text using Green's function integration. To do so, we write down the Green's functions $G_{n,m}(r,t;R,T)$ describing diffusion of molecules in $m$ dimensions released by cells in $n$ dimensions at $(R,T)$ and measured by cells at $(r,t)$. For $n \neq m$, we assume a semi-infinite environment. For cells in 1D and diffusion in 1D, we assume a narrow channel of width $h$ in both dimensions perpendicular to the channel. For cells in 1D or 2D and diffusion in 2D, we assume an extracellular medium of thickness $h$. We calculate the Green's functions for cells in two dimensions by integrating over a ring of diffusive sources at radius $R$; we calculate the Green's functions for cells in three dimensions by integrating over a shell of diffusive sources at radius $R$. Below, $I_0[.]$ is the zeroth $I$-Bessel function and $\sinh[.]$ is the hyperbolic sine function. The Green's functions are

$$G_{1,1}(r,t;R,T) = e^{-(r-R)^2/4D(t-T)}/h^2\sqrt{4\pi D(t-T)} \tag{61a}$$

$$G_{1,2}(r,t;R,T) = e^{-(r-R)^2/4D(t-T)}/2\pi h D(t-T) \tag{61b}$$

$$G_{2,2}(r,t;R,T) = R\,I_0[rR/2D(t-T)]e^{-(r^2+R^2)/4D(t-T)}/2hD(t-T) \tag{61c}$$

$$G_{2,3}(r,t;R,T) = R\,I_0[rR/2D(t-T)]e^{-(r^2+R^2)/4D(t-T)}/2\sqrt{\pi}D^{3/2}(t-T)^{3/2} \tag{61d}$$

$$G_{3,3}(r,t;R,T) = R\sinh[rR/2D(t-T)]e^{-(r^2+R^2)/4D(t-T)}/r[\pi D(t-T)]^{1/2}. \tag{61e}$$

One-by-one, we study the initiation time for these systems by studying the self-consistency relationship for the threshold concentration $C_{\text{th}}$ and initiation time $t_{\text{init}}$ for a given initial signaling colony of size $r_i$:

$$C_{\text{th}} = a\rho \int_0^{t_{\text{init}}} dT \int_0^{r_i} dR\, G_{n,m}(r_i,t_{\text{init}};R,T) = a\rho \int_0^{t_{\text{init}}} dT \int_0^{r_i} dR\, G_{n,m}(r_i,0;R,-T) \tag{62}$$

where the logic here is that the signaling wave initiates when the threshold concentration at the edge of the initial signaling colony exceeds $C_{\text{th}}$. At $t_{\text{init}}$, cells outside the colony participate in the signaling and birth a diffusive wave with dynamics we have already studied extensively. This scenario assumes the cells do not move – that the cell density is fixed. In the case of neutrophils, this means that we are ignoring the possibility that a cell initially located off the target randomly encounters the target and starts signaling. Unlike the asymptotic dynamics, the difference in diffusive and cell dimension is not the salient parameter for understanding diffusive wave initiation. Rather, the initiation dynamics are determined solely by the dimension of the diffusive environment.

We now seek to derive the equations in the main text which show the relationship between the wave initiation time $t_{\text{init}}$ and initial signaling colony size $r_i$ for various system dimensionalities when $r_i \gg D/v$ or $r_i \ll D/v$. The full relationships of $t_{\text{init}}$ versus $r_i$ for various system dimensionalities are plotted in *Figure 3* of the main text.

### Initiation with cells in 1D and diffusion in 1D

With cells and diffusion in 1D, we can perform the integral *Equation (62)* directly (for cells in 1D, we consider the bounds on the integral over $R$ to be $-r_i$ to $r_i$) and get a closed form relationship:

$$C_{\text{th}} = \frac{a\rho r_i^2}{h^2 D}\left[\left(Dt_i/\pi r_i^2\right)^{1/2} e^{-r_i^2/Dt_i} - 1 + \left(1 + Dt_i/2r_i^2\right)\text{erf}\left(r_i^2/Dt_i\right)^{1/2}\right]. \tag{63}$$

In the limit where $r_i \gg Dt_i$, we get that $t_i = 2D/v^2$ by using the asymptotic relationship *Equation (24)*. This is the minimum initiation time, $t_{\text{min},1,1}$.

$$t_{\min,1,1} = 2D/v^2 \tag{64}$$

and it tells us that $r_i \gg Dt_i$ is equivalent to $r_i \gg D/v$ – we can appeal to the natural length and time scales from our asymptotic analysis. We will soon see that this is also the minimum initiation time for cells in 2D with diffusion in 2D and cells in 3D with diffusion in 3D; this is the case because, as in the asymptotic analysis, we've essentially ignored the curvature of the target when calculating the $r_i \gg D/v$ initiation time.

In the opposite limit – $r_i \ll D/v$ – we can Taylor Expand *Equation (63)* and get

$$r_i \ll D/v : t_{\text{init}} \approx (\pi D/4v^2)(D/vr_i)^2, \tag{65}$$

thus validating our equations in the main text.

## Initiation with cells in 1D and diffusion in 2D

Next, we consider the self-consistency equation *Equation (62)* with $n = 1, m = 2$. As before, we first consider the limit of $r_i \gg Dv$ and recover (through *Equation (29)*):

$$t_{\min,1,2} = 4D/\pi v^2 \tag{66}$$

while for $r_i \ll Dv$,

$$r_i \ll D/v : \log\left(Dt_{\text{init}}/r_i^2\right) \approx 2D/vr_i. \tag{67}$$

## Initiation with cells in 2D and diffusion in 2D

Moving on, we consider the case in the main text of cells in 2D with diffusion in 2D. To perform the integration of *Equation (62)* in this case, it is easiest to rewrite the Bessel function in *Equation (61c)* in integral form, then integrate first over time. With $r_i \gg D/v$, such an analysis gives a minimum initiation time of

$$t_{\min,2,2} = 2D/v^2. \tag{68}$$

In the opposite limit of $r_i \ll D/v$,

$$r_i \ll D/v : \log(4Dt_{\text{init}}/r_i^2) \approx (2D/vr_i)^2 \tag{69}$$

as noted in the main text.

## Initiation with cells in 2D and diffusion in 3D

Now, we will see that diffusive waves do not always initiate in 3D environments. We consider the integral in *Equation (62)*, but take $t_{\text{init}} \to \infty$ which gives us a maximum concentration $C_{\max,2,3}$ at $r = r_i$ of:

$$C_{\max,2,3} = 2a\rho r_i/\pi D \tag{70}$$

Thus, by *Equation (29)*, if $r_i < D/v$, $C_{\max,2,3} < C_{\text{th}}$ and the signaling wave cannot initiate. Examining *Equation (62)* for $r_i \approx D/v$ reveals

$$r_i \approx D/v : t_{\text{init}} \approx (\pi D/16v^2)(vr_i/D)^4(vr_i/D - 1)^{-2} \tag{71}$$

while

$$t_{\min,2,3} = 4D/\pi v^2. \tag{72}$$

## Initiation with cells in 3D and diffusion in 3D

Finally, we consider initiation with cells in 3D. Again, we consider the limit $t_{\text{init}} \to \infty$ in **Equation (62)** to get:

$$C_{\text{max},3,3} = a\rho r_i^2/3D. \tag{73}$$

Thus, waves do not initiate below a critical $r_i$. However, here, the critical value is $r_i = \sqrt{3}D/v$. As with 1D cells/diffusion and 2D cells/diffusion, we recover

$$t_{\text{min},3,3} = 2D/v^2 \tag{74}$$

in the limit $r_i \gg D/v$.

## Wave initiation with discrete cells

In the main text, we claimed that the qualitative wave initiation findings hold even in systems with discrete cells. Here, we show this by considering the extreme case of a single, point-like cell at the center of an initiating colony of size $r_i$. We consider cells and diffusion in the same number of dimensions, $m$. In an $m$-dimensional diffusive environment, the concentration created by this source – which emits at a rate $a$ – at a radius $r_i$ and time $t$ will be given by:

$$c(r_i, t) = a \int_0^t dT\, e^{-r_i^2/4DT}(4\pi DT)^{-m/2}. \tag{75}$$

For $m = 3$, this integral is bounded from above by $a/4\pi D r_i$ as $t \to \infty$ while for $m = 1, 2$, the concentration diverges as $t \to \infty$. Thus we can see that, as in the continuum theory of the main text, the wave will always initiate for $m = 1, 2$ but has a critical radius for $m = 3$.

However, this is not the only qualitative similarity between the discrete and continuum cases. For $m = 1, 2$, we obtain

$$c_{m=1}(r_i, t) = \frac{a}{\sqrt{4\pi D}}\left[2\sqrt{t}e^{-r_i^2/4Dt} - r_i\sqrt{\pi/D}\,\text{erfc}\frac{r_i}{\sqrt{4Dt}}\right] \tag{76a}$$

$$c_{m=2}(r_i, t) = \frac{a}{4\pi D}\Gamma_0(r_i^2/4Dt), \tag{76b}$$

which we can manipulate to a more familiar form by considering the $r_i \ll D/v$ and $t \gg D/v^2$ limit, in which case we yield:

$$r_i \ll D/v: \; c_{m=1}(r_i, t) \approx a\sqrt{\frac{t}{\pi D}} \tag{77a}$$

$$r_i \ll D/v: \; c_{m=2}(r_i, t) \approx \frac{a}{4\pi D}\log\frac{4Dt}{r_i^2}. \tag{77b}$$

Making the substitution of $c(r_i, t) = C_{\text{th}}$, setting $t = t_{\text{init}}$, multiplying both sides of the equations by $r_i^m$, and writing $C_{\text{th}} = a\rho D/v^2 \sim aD/r_i^m$, we arrive at:

$$r_i \ll D/v, \; m = 1: \; t_{\text{init}} \sim (D/v^2)(D/vr_i)^2 \tag{78a}$$

$$r_i \ll D/v, \; m = 2: \; \log\left(Dt_{\text{init}}/r_i^2\right) \sim (Dr_i/v)^2, \tag{78b}$$

exactly the results of **Equations (65) and (69)**.

Thus, not only the rough phenomenology (of $m = 3$ being distinct from $m = 1, 2$ in that exhibits a critical radius), but also the specific scalings of initiation time hold when accounting for the discreteness of cells.

## Appendix 8

### Wave initiation with Hill function activation

We now characterize the effect of Hill function-like activation on the signaling wave initiation, as we have done already for asymptotic dynamics. To do so, we consider a simple situation: cells within a volume of size $r_i$ signal with some rate $a$ while neighboring cells outside of the initial signaling volume (i.e., with $r>r_i$) signal with a concentration-dependent rate of $ac^n/(c^n + C_{th}^n)$. These calculations give us an idea of how sensitive the initiation dynamics are to the details of cell activation. As we will show, the initiation dynamics with Hill function activation are a good approximation of the initiation dynamics for Heaviside activation when for relatively small $n$. One can imagine that such a situation may be relevant in, for example, the neutrophil swarming experiments presented in the main text (*Reátegui et al., 2017*). Here, cells in direct contact with a foreign protein begin signaling their neighbors, which respond to the presence of the signaling molecule by participating in the emission themselves. This analysis again treats the cell distribution as static and ignores the possibility that neutrophils may randomly encounter the target.

### Wave initiation with Hill function activation, cells in 1D, and diffusion in 1D

For cells in 1D with diffusion in 1D, the scenario described above can be described with the following equation of motion:

$$\frac{\partial c}{\partial t} = D\frac{\partial^2 c}{\partial r^2} + \frac{a\rho}{h^2}\Theta[r_i - |r|] + \frac{a\rho}{h^2}\Theta[|r| - r_i]\frac{c^n}{c^n + C_{th}^n}. \tag{79}$$

One can non-dimensionalize *Equation (79)* by dividing all the concentration scales by $C_{th}$, dividing all the length scales by $l_c = \sqrt{h^2 C_{th} D/a\rho}$, and dividing all the time scales by $\tau_c = h^2 C_{th}/a\rho$, thusly arriving at

$$\frac{\partial(c/C_{th})}{\partial(t/\tau_c)} = \frac{\partial^2(c/C_{th})}{\partial(r/l_c)^2} + \Theta\left[\frac{r_i - |r|}{l_c}\right] + \Theta\left[\frac{|r| - r_i}{l_c}\right]\frac{(c/C_{th})^n}{(c/C_{th})^n + 1}, \tag{80}$$

which shows that $l_c = \sqrt{h^2 C_{th} D/a\rho}$ is the relevant length scale and $\tau_c = h^2 C_{th}/a\rho$ is the relevant time scale. (In the $n \to \infty$ limit of Heaviside activation, $\tau_c = D/v^2$ and $l_c = D/v$.) As with Heaviside activation, we refer to the initiation time $t_{init}$ as the time at which $c(r_i, t_{init}) = C_{th}$. To find $t_{init}$, we numerically solve *Equation (79)* using the methods discussed in Appendix 5: Asymptotic wave dynamics with Hill function activation. This gives the relationship of $t_{init}/\tau_c$ as a function of $r_i/l_c$ and $n$ shown in *Appendix 8—figure 1A*. As seen in *Appendix 8—figure 1A*, even low-order ($n = 1, 2, 3, 5, 10$) Hill functions exhibit relatively large (compared to $\tau_c$) initiation times for $r_i \ll l_c$.

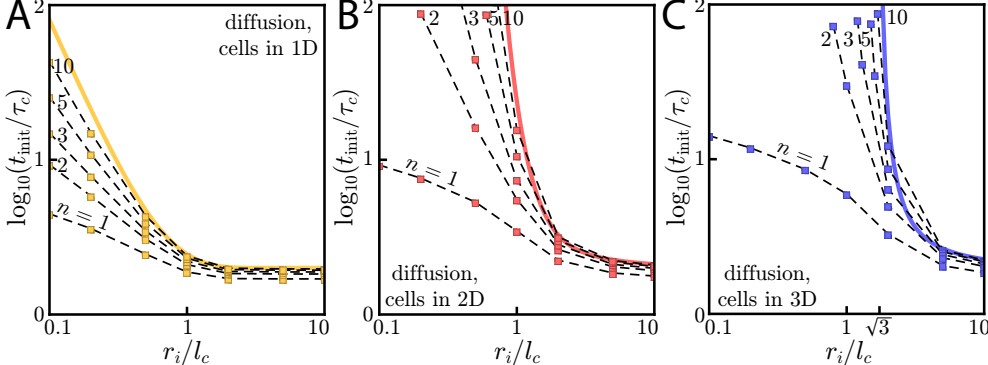

**Appendix 8—figure 1.** Diffusive wave initiation with Hill function activation. (**A**) Wave initiation for cells in 1D with diffusion in 1D. Here, the initiation time $t_{init}$ is normalized by the characteristic time scale $\tau_c = h^2 C_{th}/a\rho$ and the initial signaling colony size $r_i$ is normalized by the characteristic length

*Appendix 8—figure 1 continued on next page*

*Appendix 8—figure 1 continued*

scale $l_c = (h^2 C_{\mathrm{th}} D / a\rho)^{1/2}$. Yellow data points are individual simulations with dashed black guides to the eye connecting the yellow points. The solid yellow line corresponds to Heaviside-like activation and is reproduced from the main text. (B) Same as A, but for cells in 2D and diffusion in 2D. Here, $\tau_c = h C_{\mathrm{th}} / a\rho$ and $l_c = (h C_{\mathrm{th}} D / a\rho)^{1/2}$. (C) Same as A, but for cells in 3D and diffusion in 3D. Here, $\tau_c = C_{\mathrm{th}} / a\rho$ and $l_c = (C_{\mathrm{th}} D / a\rho)^{1/2}$. We note the value of $r_i = \sqrt{3} l_c$, the value below which waves fail to initiate for Heaviside-like activation.

## Wave initiation with Hill function activation, cells in 2D, and diffusion in 2D

Next, we study the initiation dynamics above with cells and diffusion in two dimensions. Here, the dynamics are governed by the following equation of motion:

$$\frac{\partial c}{\partial t} = D\left(\frac{\partial^2 c}{\partial r^2} + \frac{1}{r}\frac{\partial c}{\partial r}\right) + \frac{a\rho}{h}\Theta[r_i - r] + \frac{a\rho}{h}\Theta[r - r_i]\frac{c^n}{c^n + C_{\mathrm{th}}^n}. \tag{81}$$

Now that we are considering the initiation dynamics, we must include terms like $D(\partial c/\partial r)/r$, which we could previously neglect in our asymptotic analysis of cells in 2D with diffusion in 2D. The curvature of the initial signaling colony matters when calculating initiation times. Note that *Equation (81)* can be non-dimensionalized in the same spirit as *Equation (80)* with characteristic length scale $l_c = \sqrt{h C_{\mathrm{th}} D / a\rho}$ and characteristic time scale $\tau_c = h C_{\mathrm{th}} / a\rho$. As with cells and diffusion in 1D, we numerically solve *Equation (81)* using the methods discussed in Appendix 5: Asymptotic wave dynamics with Hill function activation to find $t_{\mathrm{init}}$ such that $c(r_i, t_{\mathrm{init}}) = C_{\mathrm{th}}$. Here again, we see that even low-order ($n = 2, 3, 5, 10$) Hill functions exhibit relatively large (compared to $\tau_c$) initiation times for $r_i \ll l_c$.

## Wave initiation with Hill function activation, cells in 3D, and diffusion in 3D

Finally, we study signaling wave initiation properties of a 3D environment by studying wave initiation with cells and diffusion in 3D. To do so, we numerically solve the 3D analog of *Equation (81)*,

$$\frac{\partial c}{\partial t} = D\left(\frac{\partial^2 c}{\partial r^2} + \frac{2}{r}\frac{\partial c}{\partial r}\right) + a\rho\Theta[r_i - r] + a\rho\Theta[r - r_i]\frac{c^n}{c^n + C_{\mathrm{th}}^n}, \tag{82}$$

which can be non-dimensionalized in the same spirit as *Equation (80)*, but with characteristic length scale $l_c = \sqrt{C_{\mathrm{th}} D / a\rho}$ and characteristic time scale $\tau_c = C_{\mathrm{th}} / a\rho$. We solve *Equation (82)* using the methods discussed in Appendix 5: Asymptotic wave dynamics with Hill function activation to find $t_{\mathrm{init}}$ such that $c(r_i, t_{\mathrm{init}}) = C_{\mathrm{th}}$. This gives the numerically determined relationship of $t_{\mathrm{init}}$ as a function of $r_i$ and $n$ shown in *Appendix 8—figure 1C*. We see that even low-order ($n = 2, 3$) Hill functions exhibit relatively large (compared to $\tau_c$) initiation times for $r_i \ll l_c$. Larger yet Hill functions ($n = 5, 10$) can lead to very large initiation times (compared to $\tau_c$) even for $r_i \approx l_c$.

In fact, $n > 3$ activation functions can result in initiation failures when $a\rho r_i^2 / 3 D C_{\mathrm{th}} \ll 1$. To see that this is the case, we treat *Equation (82)* in the steady state ($\partial c/\partial t = 0$) and use a perturbative analysis, assuming $c^n/(c^n + C_{\mathrm{th}}^n) \ll 1$, in which case $c^n/(c^n + C_{\mathrm{th}}^n) \approx (c/C_{\mathrm{th}})^n$. In such a situation, we can write $c(r) \approx c_0(r) + c_1(r)$ as the sum of a dominant contribution $c_0$ that is generated by cells within $r_i$ and satisfies

$$0 = D\left(\frac{\partial^2 c_0}{\partial r^2} + \frac{2}{r}\frac{\partial c_0}{\partial r}\right) + a\rho\Theta[r_i - r] \tag{83}$$

and a small correction $c_1$ that is generated by cells beyond $r_i$ and obeys

$$0 = D\left(\frac{\partial^2 c_1}{\partial r^2} + \frac{2}{r}\frac{\partial c_1}{\partial r}\right) + a\rho\frac{c_0^n}{c_0^n + C_{\text{th}}^n}\Theta[r - r_i] \approx D\left(\frac{\partial^2 c_1}{\partial r^2} + \frac{2}{r}\frac{\partial c_1}{\partial r}\right) + a\rho\left(\frac{c_0}{C_{\text{th}}}\right)^n\Theta[r - r_i]. \tag{84}$$

We can solve *Equation (83)* directly to arrive at:

$$c_0(r<r_i) = \frac{a\rho}{2D}\left(r_i^2 - r^2/3\right),\ c_0(r>r_i) = \frac{a\rho r_i^3}{3Dr} \tag{85}$$

where the form of $c_0(r>r_i)$ is reminiscent of solving for the potential of a uniformly charged sphere in electrostatics (*Berg and Purcell, 1977*). If $a\rho r_i^2/3DC_{\text{th}} = \epsilon \ll 1$, we can calculate the perturbation $c_1$. By *Equation (84)*, $c_1$ obeys

$$0 = D\left(\frac{\partial^2 c_1}{\partial r^2} + \frac{2}{r}\frac{\partial c_1}{\partial r}\right) + a\rho\left(\frac{c_0}{C_{\text{th}}}\right)^n\Theta[r - r_i] = D\left(\frac{\partial^2 c_1}{\partial r^2} + \frac{2}{r}\frac{\partial c_1}{\partial r}\right) + a\rho\left(\frac{\epsilon r_i}{r}\right)^n\Theta[r - r_i] \tag{86}$$

so that

$$c_1(r<r_i) = \frac{a\rho r_i^2\epsilon^n}{D(n-2)},\ c_1(r>r_i) = \frac{a\rho\epsilon^n}{D(n-3)}\left[\frac{r_i^3}{Dr} + \frac{r^2}{n-2}\left(\frac{r_i}{r}\right)^n\right]. \tag{87}$$

For $c_1$ to be a sensible perturbative correction, we require it to be positive (since we are adding source terms to $c_0$ to get $c_1$) and much smaller than $c_0$. This is the case when $n>3$. In this limit, it is smaller than $c_0$ by roughly a factor of $\epsilon^n$ – a very small correction. Thus, $n>3$ activation functions can give steady-state concentration profiles that do not trigger waves in a three-dimensional diffusive environment.

In *Appendix 8—figure 1C*, we can indeed see that small $n$ activation functions show less appreciable increases in the initiation time as $r_i$ decreases.

## Appendix 9

### Sensitivity of the information front to fit parameters

As an example of the Green's function method described in Materials and methods, we have plotted various information wave fronts in *Appendix 9—figure 1*. These wave fronts assume different values of the diffusion constant $D$ and the threshold concentration $C_{\text{th}}$ is fit to give the experimentally observed wave initiation time in *Reátegui et al., 2017*. As we can see from the plot, there is a small range of values for $D$ for which one can construct an information wave front that agrees with the data (*Appendix 9—figure 1 A*). These values of $D$ are consistent with the diffusion constant of small molecules like LTB4. Values of $D$ differing significantly from this range give information wave fronts that differ significantly from the experimentally observed wave front.

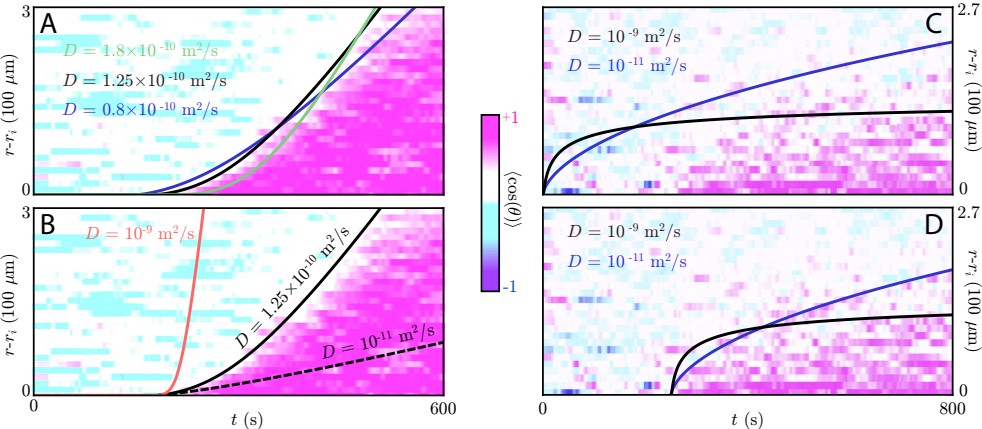

**Appendix 9—figure 1.** Wavefront fitting of the simple diffusion and relay models. (**A**) Plot showing the information waves for various choices of diffusion constant (black) overlaid on the experimental chemotactic index data from *Reátegui et al., 2017* (color plot). Here, we take $r_i = 100 \ \mu$m, although the target in the experiment is a smaller, oblong object. The size of the target has no effect on the convex shape of the information wave front. The black line is reproduced from the main text and has $D = 1.25 \times 10^{-10}$ m$^2$/s and asymptotic wave speed $v \approx 1.7 \ \mu$m/s (the threshold concentration is $C_{\text{th}}/a\rho = 2/\pi v$). This diffusion constant is consistent with a small molecule like LTB4, and the resulting information wave dynamics can be made to fit the information wave front – both the initiation time of $\approx 200$ s and the transient dynamics. Other choices of parameters (green: $D = 1.8 \times 10^{-10}$ m$^2$/s, $v \approx 2.3 \ \mu$m/s, navy: $D = 0.8 \times 10^{-10}$ m$^2$/s, $v \approx 1.3 \ \mu$m/s) give information wave fronts that are also roughly consistent with the experimental data. (**B**) However, with a much larger ($D = 10^{-9}$ m$^2$/s, red) or smaller ($D = 10^{-11}$ m$^2$/s, dashed) diffusion constant, an information wave with the correct initiation time does not have the correct transient dynamics. The wave speeds for these larger and smaller diffusion constants are, respectively, $v \approx 11 \ \mu$m/s and $v \approx 0.3 \ \mu$m/s. (**C**) Simple diffusion models of various diffusion constants overlaid atop data from Reategui et al. in which the the same swarming assay as in A/B (but with a slightly smaller 80 μm target) was utilized but with neutrophils whose LTB4 receptors (BLT1/2) had been blocked. Here, we see a 250 s offset before the propagation of a slow-moving diffusive front. A simple diffusive model does not capture this offset well, but does accurately capture the concave shape of the front (which contrasts with the convex shape of the front propagated by a relay). (**D**) Same as in C but with a 250 s offset in the theoretical curves. These curves fit the observed chemotactic index dynamics fairly well.

### Physiological relevance of these fit parameters

With the fit values of $C_{\text{th}}/a\rho = 3.67 \times 10^5$ s/m and $D = 1.25 \times 10^{-10}$ m$^2$/s reported in the main text and the empirical cell density of $\rho = (1/50 \ \mu\text{m})^2$, we conclude that $C_{\text{th}}/a \approx 1.5 \times 10^{14}$ s/m$^3$.

Under similar conditions with the swarming assay, (*Reátegui et al., 2017*) report neutrophil LTB4 production in a 3 mm thick extracellular medium to be approximately 1 pg per 100 microliters per hour. With cells at a density of $\rho = (1/50 \ \mu\text{m})^2$ and a 3 mm thick extracellular medium, this

corresponds to a production rate of $a \approx 40$ molecules/second/cell. Thus, combining this production rate with the above, we yield $C_{\text{th}} \approx 500$ pM.

This value is within the range of the measured BLT1 receptor affinity for LTB4, which is reported to be approximately $0.1 - 2$ nM (*Yokomizo, 2015*).

## Appendix 10

### Simple diffusion model

In the main text, we showed the qualitative differences between a signaling relay, in which cells emit one type of signaling molecule in response to the local concentration of the same molecule, and a simple diffusive signaling model, in which cells within some volume signal surrounding cells, which do not participate in the signaling at all. Here, we explicitly calculate some of the properties of a simple diffusion model. We consider cells in 2D, with a region of cells of size $r_i$ at $z = 0$ in which the cells emit diffusible signaling molecules at rate $a$. In equation form,

$$\frac{\partial c}{\partial t} = D\left(\frac{\partial^2 c}{\partial r^2} + \frac{1}{r}\frac{\partial c}{\partial r} + \frac{\partial^2 c}{\partial z^2}\right) + a\rho\delta(z)\Theta[r_i - r] \tag{88}$$

describes the concentration in space and time. To calculate concentrations, one can either propagate this equation directly or, as we do in the main text, integrate Green's functions in a manner similar to that described above. For cells in 2D with diffusion in 3D (assuming a semi-infinite environment), this gives

$$c(r, z = 0, t) = a\rho \int_0^t dT \int_0^{r_i} dR\, G_{2,3}(r, 0; R, -T) \tag{89}$$

as the concentration at $z = 0$. One can take gradients according to $\partial c/\partial r$.

In **Appendix 9—figure 1**, we show that these dynamics produce information wave fronts that are different from the information wave fronts of diffusive relays. Moreover, the information wave fronts of simple diffusive theories are inconsistent (**Appendix 9—figure 1C**) with the observed chemotactic index dynamics catalogued by **Reátegui et al., 2017** for neutrophils with blocked LTB4 (BLT1/2) receptors. With a time delay of ~250 s, the information wave fronts of simple diffusive models can be made consistent with the observed information wave fronts. As there is great ambiguity about the signaling molecules that govern recruitment of BLT1/2-blocked neutrophils, we have shown this is true for a large range (two orders of magnitude) of diffusion constants.

In the limit $r \gg r_i$, the signaling colony looks like a point source, meaning that **Equation (89)** can be simplified according to $\int dR\, G_{2,3} \approx r_i^2 e^{-r^2/4DT}/(4\sqrt{\pi}D^{3/2}T^{3/2})$. Thus, we get that

$$r \gg r_i : \; c(r, z = 0, t) \approx \frac{a\rho r_i^2}{2\sqrt{\pi}D^{3/2}} \int_0^t dT\, \frac{e^{-r^2/4DT}}{T^{3/2}} = \frac{a\rho r_i^2}{2Dr}\mathrm{erfc}\left(\frac{r^2}{4Dt}\right)^{1/2} \tag{90}$$

which, in the limit of $r^2 \gg Dt$, gives

$$r \gg r_i, \sqrt{Dt} : \; c(r, z = 0, t) \approx \frac{a\rho r_i^2}{Dr}\mathrm{erfc}\left(\frac{r^2}{4Dt}\right)^{1/2} \approx \frac{a\rho r_i^2}{r^2}\sqrt{\frac{t}{\pi D}}\, e^{-r^2/4Dt} \tag{91}$$

which shows that the concentration profiles of simple diffusive models indeed have very shallow, Gaussian (with $1/r^2$ adjustments) tails.

Similarly, for cells in 2D with diffusion in 2D,

$$c(r, t) = \frac{a\rho}{h} \int_0^t dT \int_0^{r_i} dR\, G_{2,2}(r, 0; R, -T) \tag{92}$$

describes the concentrations. In the same limits ($r \gg r_i, \sqrt{Dt}$), we get

$$r \gg r_i, \sqrt{Dt} : \; c(r, t) \approx \frac{a\rho r_i^2 t}{hr^2} e^{-r^2/4Dt} \tag{93}$$

as the concentration. Here again, we see that the concentration profiles are shallow Gaussian with $1/r^2$ adjustments. We plot the resulting gradients in this thin extracellular medium limit against the gradients from a comparable relay model in **Appendix 10—figure 1B**.

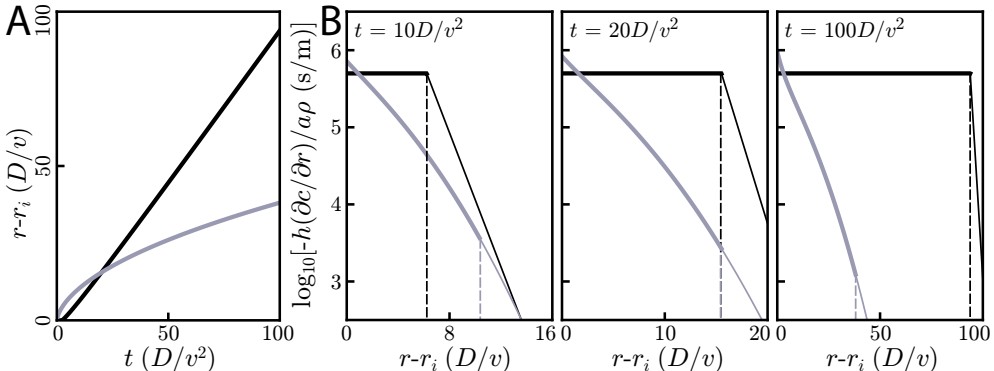

**Appendix 10—figure 1.** Comparison of relay and simple diffusion for cells and diffusion in 2D. All plots are for $D = 10^{-10}$ m$^2$/s, $v = 2$ $\mu$m/s, and $r_i = 4D/v$. (**A**) Information fronts for relay (black) and simple diffusion (gray) models. The information wave travels like $\sqrt{t}$ for simple diffusion and $vt$ for the relay. The threshold concentration for the simple diffusion model is chosen such that the its information front intersects the relay model at $t = 20D/v^2$. Thus, $hC_{\text{th,relay}}/a\rho = 25$ s while $hC_{\text{th,diff.}}/a\rho \approx 0.25$ s. (**B**) Snapshots of radial gradients at various times for both the relay (black) and simple diffusion (gray) models. The dashed vertical lines indicate the location of the wave fronts. For short times (left) the relay's information wave front lags behind the simple diffusion model's information front, though it later catches up (middle) and passes it (right). At all times, for cells just inside the wave front, the relay model creates gradients that are orders of magnitude larger than does simple diffusion.

## Appendix 11

### Quantifying the effects of chemotaxis

To understand the effects that chemotaxis has on our model, we consider cells in 2D. The results below can be adopted to study cells in 3D or 1D, although the dimensionality of the cells has no effect on the asymptotic signaling wave speed.

We consider the same signaling motif as in the main text – that cells emit a diffusible molecule with rate $a$ once the local concentration of the same molecule exceeds $C_{th}$ – but now consider a time-varying density. Our model is a coarse-grained one; we study the case of cells moving toward the origin (radially inward) with a mean speed $u$ once the local concentration exceeds $C_{th}$. This is a toy model of neutrophil chemotaxis and is, of course, an approximation because the mean radial speed will depend on – among many other factors – the strengths of the gradients the cells use for chemotaxis. In full, for cells in any number of dimensions and within this model,

$$\frac{\partial c(\mathbf{r},t)}{\partial t} = D\nabla^2 c + a\rho(\mathbf{r},t)\Theta[c - C_{th}] \tag{94a}$$

$$\frac{\partial \rho(\mathbf{r},t)}{\partial t} = \nabla \cdot (u\hat{r}\rho)\Theta[c(\mathbf{r},t) - C_{th}] \tag{94b}$$

where $\hat{r}$ is the unit vector pointing radially outward. For cells in 2D,

$$\frac{\partial c(r,z,t)}{\partial t} = D\left(\frac{\partial^2 c}{\partial r^2} + \frac{1}{r}\frac{\partial c}{\partial r} + \frac{\partial^2 c}{\partial z^2}\right) + a\rho(r,t)\delta(z)(\Theta[c - C_{th}]\Theta[r - r_i] + \Theta[r_i - r]) \tag{95a}$$

$$\frac{\partial \rho(r,t)}{\partial t} = \frac{u}{r}\frac{\partial(r\rho)}{\partial r}\Theta[c - C_{th}]\Theta[r - r_i] \tag{95b}$$

which are the coupled equations we will study going forward. Note that we have included a source term for cells within the initial signaling colony of radius $r_i$.

We again assume a signaling wave propagates at outward with speed $v$. In this case, the cell density beyond the target is described by:

$$\rho(r_i < r < vt, t) = \rho_0 \frac{1 + ut/r}{(1 + u/v)^2} \tag{96}$$

which one can derive by assuming an outward propagating wave with speed $v$, a group of inward chemotaxing cells with speed $u$, and an initially uniform density of cells $\rho_0$. To do so, we consider the signaling wave passing a cell at radius $R$ and time $t - T$. At a later time $t$, the cell initially at $R$ has moved inward a distance $uT$. Thus, the density at $r = R - uT$ and $t$ is $\rho(r,t) \sim \rho_0 R/r = \rho_0(1 + ut/r)/(1 + u/v)$. Integrating this density and demanding conservation of cell number gives *Equation (96)*.

### Effect on asymptotic wave speed relationships

Before numerically solving *Equations (95a)* and (95b) to show the precise effects chemotaxis has on the concentration profiles, concentration profiles, and information wave fronts, we calculate the effect it has on the asymptotic wave speed.

To do so, we first show that only cells within $\approx D/v$ of the wave front contribute to the concentration at the wave front. To contribute to the concentration at the wave front, you need to be within about a diffusion length of it. If the wave front passed a time $t$ ago, that means being within $\delta r \approx \sqrt{Dt}$. However, we know that $t = \delta r/v$, so $\delta r \approx D/v$ – the characteristic length scale of diffusive waves.

As only cells within $\approx D/v$ of the wave front contribute to concentration at the wave front, we are considering cell densities on the order of:

$$\rho(r = vt - D/v, t) = \rho_0 \frac{1 + ut/r}{(1 + u/v)^2} \approx \rho_0 \frac{1 + ut/(vt - D/v)}{(1 + u/v)^2}. \tag{97}$$

In the asymptotic regime of $vt \gg D/v$, we get

$$\rho \approx \rho_0/(1 + u/v) \tag{98}$$

meaning the density of cells that contribute to the wave front propagation is approximately constant. Therefore, we may modify the analysis that lead to *Equations (24) and (29)* to get two new asymptotic equations for the wave speed:

$$h \ll D/v: \ C_{\text{th}} = \frac{a\rho_0 D}{hv^2(1 + u/v)} \tag{99}$$

and

$$h \gg D/v: \ C_{\text{th}} = \frac{2a\rho_0}{\pi v(1 + u/v)}. \tag{100}$$

For neutrophils and the information wave front presented in the main text (reproduced in *Appendix 11—figure 1*), $1 + u/v \approx 1 + (0.3\ \mu\text{m/s})/(2\ \mu\text{m/s}) = 1.15$ and the effect of chemotaxis on the asymptotic dynamics is small.

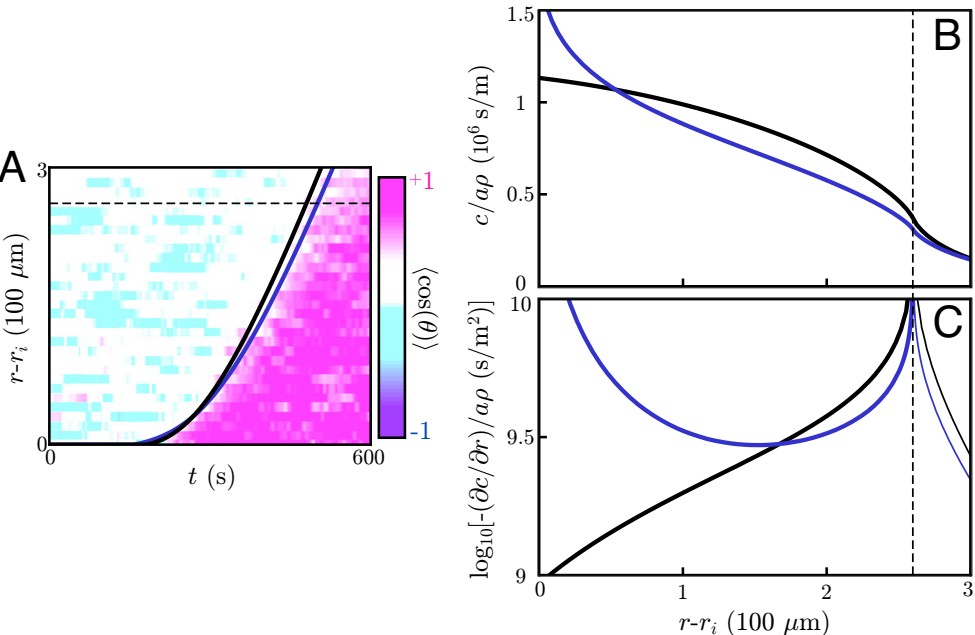

**Appendix 11—figure 1.** Effects of chemotaxis on wavefront location and concentration profiles. (**A**) Information wave fronts for cell signaling relays with (navy) and without (black) chemotaxis. The information wave fronts are overlaid on the experimental chemotactic index data from (color plot) (*Reátegui et al., 2017*). The black curve is reproduced from the main text. Both models can account for the observed information wave fronts by fitting two parameters: the signaling molecule diffusivity, $D$ and the threshold concentration, $C_{\text{th}}$. (**B/C**) Concentration profiles (**B**) and gradients (**C**) generated by the signaling relay models in A. The wave front is indicated in all panels by the dashed line, and the concentration profiles and gradients are plotted at the times such that the threshold concentration is equal to the concentration at the wave front. When one accounts for chemotaxis, the concentration profiles near the target steepen relative to models without chemotaxis.

## Effect on transient dynamics

To study the effect of chemotaxis on the transient dynamics of neutrophil swarming, we utilize a modified version of our Green's function method reported in Materials and methods to propagate *Equations (95a)* and *Equations (95b)*. One can do so using the same algorithm described previously, but with the following modifications with $r_c(t)$ the information wave front:

- For radii $r$ at time $t$ satisfying $r_i < r < r_c(t)$, find the time $t^*(r, t)$ at which the neutrophils at radius $r$ at time $t$ began chemotaxing inward. This time satisfies the relationship $r_c[t^*] - r = u(t - t^*)$.
- The density at $r$ is therefore given by the ratio of $r_c[t^*]$ to $r$ with an additional factor of one plus the ratio of the inward advection speed to the outward-propagating wave speed at $r_c[t^*]$ and $t^*$:

$$\rho(r, t) = \rho_0 \frac{r_c[t^*]/r}{1 + u\left(\frac{\partial r_c[t^*]}{\partial t}\right)^{-1}}.$$

(101)

This is analogous to *Equation (96)* and reduces to *Equation (96)* in the asymptotic limit of $r_c(t) = vt$.

- We assume that once the cells reach the target edge, they pack inward at a maximum density, in units of the cell diameter $d_c$, of $\rho_{\max} = 1/d_c^2$. For the experiments we discuss (*Reátegui et al., 2017*), this means that $\rho_{\max} \approx 10\, \rho_0$.

With all these adjustments, and using the reported value of $u \approx 20$ $\mu$m/s in *Reátegui et al., 2017*, we arrive at the navy information wave front in *Appendix 11—figure 1A*. This curve is a fit by eye to the experimental information wave front and has fit parameters of $D = 1.5 \times 10^{-10}$ m$^2$/s and $v = 1.73$ $\mu$m/s, corresponding to a threshold concentration of $C_{\text{th}}/a\rho_0 = \frac{2}{\pi v(1+u/v)} \approx 3.07 \times 10^5$ s/m. For reference, the black curve in *Appendix 11—figure 1A* is the information wave front from *Figure 4* of the main text, for which $D = 1.25 \times 10^{-10}$ m$^2$/s and $C_{\text{th}}/a\rho_0 = \frac{2}{\pi v(1+u/v)} \approx 3.67 \times 10^5$ s/m. Thus, including chemotaxis only negligibly affects our fit values.

To compare the two models in *Appendix 11—figure 1A*, we plot both the concentration profile (*Appendix 11—figure 1B*) and the concentration gradient (*Appendix 11—figure 1C*) for a given critical radius (the dashed line in *Appendix 11—figure 1A*). When one accounts for chemotaxis, the concentration profile near the target steepens relative to a model with a stationary cell distribution. We can therefore see that chemotaxis itself can lead to steeper concentration profiles, though the model we have explored here only accounts for an average inward drift of the cells.

