## [Decision Letter]

**Acceptance summary:**

The manuscript by Dieterle, et al. provides a theoretical framework for understanding the initiation and propagation of diffusive relay signaling waves. While the analysis of such waves is extensively available in prior literature the novelty of the current work lies in its focus on the role of dimensionality for the diffusing substance and the releasing sources.

**Decision letter after peer review:**

Thank you for submitting your article "Dynamics of diffusive cell signaling relays" for consideration by *eLife*. Your article has been reviewed by two peer reviewers, and the evaluation has been overseen by a Reviewing Editor and Aleksandra Walczak as the Senior Editor. The following individuals involved in review of your submission have agreed to reveal their identity: Elena F Koslover (Reviewer #2); James Parkin (Reviewer #3).

The reviewers have discussed the reviews with one another and the Reviewing Editor has drafted this decision to help you prepare a revised submission.

Summary:

The manuscript by Dieterle, et al. provides an excellent theoretical framework for understanding the initiation and propagation of diffusive relay signaling waves. While the analysis of such waves is extensively available in prior literature (including both early mathematical analyses from half a century ago and much more recent studies applying these models to specific biological measurements), the novelty of the current work lies in its focus on the role of dimensionality for the diffusing substance and the releasing sources. Cell-cell signaling systems are rarely modeled using a single species, however, their results are made widely applicable by (1) identifying when biological and biophysical details are irrelevant to wave behavior, and (2) extending derivations to common constraints such as decay, pulse width, chemotaxis, and a Hill-like source term.

The main text presents largely straight-forward arguments based on dimensional analysis, and more explicit mathematical derivations are included in the supplementary material. Universal behaviors are clearly indicated, highlighting how important features of these propagating waves are independent of the signaling mechanism details. The logic of the manuscript is beautifully laid out, making it easy to follow for the non-expert, and the final section connecting to experimental data on neutrophil swarming makes an excellent capstone demonstrating the practical applicability of the theoretical results.

The relationships derived here are relevant, informative, and are improvements over the scaling laws typically referenced in the cell-cell signaling literature. The supplementary information is a thorough and complete documentation of the arguments and conclusions included in the main text.

Essential revisions:

1) It may be helpful to touch upon in the Discussion how the morphology of a cellular surface would affect wave propagation or initiation in the case of 2D cells and 3D diffusion. Specifically, if cells are not on a flat plane but on a highly convoluted surface (as seen in some epithelia, for instance), or if they are on the boundary of a relatively wide tube, how would the results be affected and would you expect a transition in behavior from one effective dimensionality to another? We are not asking for a complete analysis here, but rather a concise qualitative discussion would be helpful on how bending of the cell surface might affect results, for instance by considering the extreme case of two parallel layers of cells with diffusion between them and commenting on when the system is expected to transition from effectively 3D to effectively 2D.

2) A discussion of how discrete or well-separated cells affect wave propagation is provided in the Appendix. It would be helpful to also include a discussion of how discrete or scattered cells would be expected to affect specifically wave initiation. In particular, what is the interplay between the size of the initiating colony and its separation from its neighbors in determining whether a wave can successfully initiate?

3) The manuscript does not seem to provide any description or details of the numerical simulations used to validate the approximate analytical results in Figures 1-2. A description of how these were carried out should be included in the supplementary information or a Materials and methods section.

4) The application of the theoretical results to the experimental data validates only that the neutrophil signaling relay produces a front which travels faster than diffusion, a fact that can be reasonably deduced from the heatmap in Figure 4B by eye, and an approximation of the diffusivity of LTB4. The referenced data source includes an inhibited relay experiment, more data from the heatmap shown in Figure 4, and quantification of the signaling molecule concentrations at different time points. We are curious about whether parameters that fit data from the simple diffusion match those that the authors fit to the signal relay demonstrations. Does the response profile appear consistent as the wave moves? Do the measurements of LTB4 in the reference match the expected production according to the model? Based on the fitted parameters, can the authors predict the behavior of the system in different diffusive environments?

5) Including the neutrophil swarming data in the main text succeeds in its purpose, which is to motivate the formal analysis and to help illustrate the benefits of relay propagation over simple diffusion that the immune system might exploit. However, we see no reason why the comparisons mentioned above should not be included in the supplement when the data is available.

---

## [Author Response]

Essential revisions:1) It may be helpful to touch upon in the Discussion how the morphology of a cellular surface would affect wave propagation or initiation in the case of 2D cells and 3D diffusion. Specifically, if cells are not on a flat plane but on a highly convoluted surface (as seen in some epithelia, for instance), or if they are on the boundary of a relatively wide tube, how would the results be affected and would you expect a transition in behavior from one effective dimensionality to another? We are not asking for a complete analysis here, but rather a concise qualitative discussion would be helpful on how bending of the cell surface might affect results, for instance by considering the extreme case of two parallel layers of cells with diffusion between them and commenting on when the system is expected to transition from effectively 3D to effectively 2D.

We thank the reviewers for noticing this gap in our Discussion. We have addressed it in a paragraph between Equations 7 and 8. There, we note that:

“The same dynamics hold for cells on a curved surface (such as epithelia) as long as the length scale of the curvature and the thickness of the extracellular medium are both large compared to D/v. [….] Similarly, cells on the surface of a tube with diffusion in the tube’s interior will interpolate between these two limits: when the radius of the tube is large compared to D/v, the dynamics will be of cells in 2D and diffusion in 3D; when the radius of the tube is small compared to D/v, the dynamics will be of diffusion and cells in 1D.”

2) A discussion of how discrete or well-separated cells affect wave propagation is provided in the Appendix. It would be helpful to also include a discussion of how discrete or scattered cells would be expected to affect specifically wave initiation. In particular, what is the interplay between the size of the initiating colony and its separation from its neighbors in determining whether a wave can successfully initiate?

This important distinction has been clarified in the main text (right after Equation 12) and again in Appendix 7. In these sections, we analyze the extreme limit of discreteness: initiation by one point-like cell on a lattice. This analysis shows that the fundamental role of dimensionality discussed in the main text – that systems with diffusion in 1D and 2D always have a finite trigger time, but that 3D systems do not – is preserved in discrete systems. In other words, wave initiation is qualitatively unaffected with discrete sources, but there is a quantitative adjustment that can be calculated approximately (see Appendix 7) in this limit.

We have also added two references: one to a paper by Keener and one to an in-preparation manuscript. The former very clearly treats the effects of discrete sources on the propagation of waves in one dimension; the latter more completely handles the dimension-dependent effects of discrete sources.

3) The manuscript does not seem to provide any description or details of the numerical simulations used to validate the approximate analytical results in Figures 1-2. A description of how these were carried out should be included in the supplementary information or a Materials and methods section.

We had previously provided a description of our numerical method for information wave front finding in the subsection “Green’s function integration for information wave front finding” of the supplementary information (now in the Materials and methods). We have added some detail to this text and have also referenced it explicitly in the main text (see Figures 1/2 legends, the text after Equation 6, and the text after Equation 9) and uploaded the relevant code to github.

4) The application of the theoretical results to the experimental data validates only that the neutrophil signaling relay produces a front which travels faster than diffusion, a fact that can be reasonably deduced from the heatmap in Figure 4B by eye, and an approximation of the diffusivity of LTB4. The referenced data source includes an inhibited relay experiment, more data from the heatmap shown in Figure 4, and quantification of the signaling molecule concentrations at different time points. We are curious about whether parameters that fit data from the simple diffusion match those that the authors fit to the signal relay demonstrations. Does the response profile appear consistent as the wave moves? Do the measurements of LTB4 in the reference match the expected production according to the model? Based on the fitted parameters, can the authors predict the behavior of the system in different diffusive environments?5) Including the neutrophil swarming data in the main text succeeds in its purpose, which is to motivate the formal analysis and to help illustrate the benefits of relay propagation over simple diffusion that the immune system might exploit. However, we see no reason why the comparisons mentioned above should not be included in the supplement when the data is available.

We agree with the reviewers that additional information related to the experimental demonstrations in Reategui et al. will help to orient readers.

Consequently, we have included in the appendices (see Appendix 9—figure 1C/D – these panels are new – and accompanying text) an analysis of the simple diffusion model applied to experiments from Reategui et al. in which LTB4 receptors have been blocked. What this analysis demonstrates is that simple diffusion models cannot explain the chemotactic index response observed in these experiments. However, the observed chemotactic index response is consistent with a large number of simple diffusion models (i.e., over a large range of the diffusion constant, D) in which cells have a ~250 second delay prior to the emission of some signaling molecule. Of special note – now mentioned explicitly in the main text and Appendix 10 – is the fact that these experiments show a concave chemotactic index profile, as distinct from the convex profile observed in the relay (main text and Appendix 9—figure 1A/B). It should not go without saying that there are dozens of candidate signaling molecules which govern accumulation of neutrophils and short-range recruitment, as outlined in Reategui et al. Whereas a large range of diffusion constants seem to account for the dynamics of the cells with blocked LTB4 receptors, only a narrow range of diffusion constants match the relay dynamics to theory, indicating that those dynamics are indeed governed by one signaling molecule.

We have also included – briefly in the main text, more substantively in Appendix 9 – a calculation that helps to substantiate our fitting parameters for the relay model in Figure 4 of the main text. We will outline the calculations here:

1) First, we use the fit values of the normalized (by emission rate and cell density) threshold concentration and the cell density reported in the experiments (one cell every [50 microns]^2) to gain a numerical value for Cth/a = 1.5x10^14 s/m^3.

2) Next, we use the empirically measured LTB4 concentration (see Reategui et al. Figure 4B) of neutrophils at comparable density over a long time period (60 minutes) to ascertain that the LTB4 release rate is approximately 40 molecules per cell per second.

3) Combining (1) and (2), we may deduce that Cth is approximately 500 pM.

4) This concentration is close to the 0.1-2 pM reported affinity of the BLT1 receptor for LTB4. (The BLT2 receptor has a much higher ~20 nM affinity, which may help the neutrophils deal with saturation effects as they climb the steep LTB4 gradients discussed in this work.)

It should be noted that the above calculations are very rough and rely on imprecise inference of parameters since the experiment in (1) utilized a different patient sample from the experiment in (2).

We are currently working toward more accurate, reproducible measurements of these and other quantities mentioned by the reviewers, including the consistency of the chemotactic response, the predictive value of the relay in one diffusive environment given measured parameters from another, etc. Through this ongoing set of experiments, we have repeatedly observed the coarse experimental conclusions discussed in this paper (a wave-like chemotactic index response, a chemotactic index response inconsistent with simple diffusion, a sensitive dependence of the initiation time on the size of the initial colony), with more precise measurements of the neutrophil response and more rigorous tests of the predictive power of our model forthcoming.